**Understanding the Australian Monsoon change during the Last Glacial Maximum with multi-model ensemble**

**Mi Yan[1,2], Bin Wang[3,4], Jian Liu[1,2*], Axing Zhu[1,2,5,6], Liang Ning[1,2,7] and Jian Cao[4]**

Key Laboratory of Virtual Geographic Environment, Ministry of Education; State key Laboratory of Geographical Environment Evolution, Jiangsu Provincial Cultivation Base; School of Geography Science, Nanjing Normal University, Nanjing, 210023, China.
Jiangsu Center for Collaborative Innovation in Geographical Information Resource Development and Application, Nanjing, 210023, China.
Department of Atmospheric Sciences, University of Hawaii at Manoa, Honolulu, HI 96825, USA.
Earth System Modeling Center, Nanjing University of Information Science and Technology, Nanjing, 210044, China.
State Key Lab of Resources and Environmental Information System, Institute of Geographic Sciences and Natural Resources Research, Chinese Academy of Sciences, Beijing, 100101, China
Department of Geography, University of Wisconsin-Madison, Madison, WI 53706, USA
Climate System Research Center, Department of Geosciences, University of Massachusetts, Amherst, 01003, USA.

* Corresponding author address: Dr. Jian Liu, School of Geography Science, Nanjing Normal University, 1 Wenyuan Road, Nanjing 210023, China

E-mail: jliu@njnu.edu.cn

**Abstract**

The response of Australian monsoon to the external forcings and the related mechanisms during the Last Glacial Maximum (LGM) is investigated by multi-model experiments in CMIP5/PMIP3. Although the annual mean precipitation over the Australian monsoon region decreases, the annual range, or the monsoonality, is enhanced. The precipitation increases in early austral summer and decreases in austral winter, resulting in the amplified annual range, but the main contribution comes from the decreased precipitation in austral winter. The decreased winter precipitation is primarily caused by weakened upward motion, although reduced water vapor has also a moderate contribution. The weakened upward motion is induced by the enhanced land–sea thermal contrast, which intensifies the divergence over northern Australia. The increased Australian monsoon rainfall in early summer, on the other hand, is an integrated result of the positive effect of local dynamic processes (enhanced moisture convergence) and the negative effect of thermodynamics (reduced moisture content). The enhanced moisture convergence is caused by two factors: the strengthened northwest–southeast thermal contrast between the cooler Indochina–western Indonesia and the warmer northeastern Australia, and the east–west sea surface temperature gradients between the warmer western Pacific and cooler eastern Indian Ocean, both due to the alteration of land–sea configuration arising from the sea level drop. The enhanced Australian monsoonality in the LGM is not associated with global scale circulation change such as the shift of the ITCZ, rather, it is mainly due to the change of regional circulations around Australia arising from the changes in land-sea contrast and the east-west SST gradients over the Indo-western Pacific oceans. This finding should be taken into account when investigating its future change under global warming. Our findings may also explain why proxy records indicate different changes in Australian monsoon precipitation during the LGM.

## 1 Introduction

The changes of the Australian monsoon are crucial for human society and ecology in Australia (Reeves et al., 2013a), considering the socio-economic importance of monsoon rainfall (Wang et al., 2017). As the monsoons of the summer hemisphere are linked via outflows from the opposing winter hemisphere, the Australian monsoon can also influence the Asian–Indonesian–Australian monsoon system (Eroglu et al., 2016). It is important to understand how and why the Australian monsoon would change in response to global climate change.

With strong climatic forcings (including low greenhouse gas (GHG) concentrations, large ice-sheets, and low sea level, etc.), the Last Glacial Maximum (LGM) is one of the key periods that provides an opportunity to better understand the mechanisms of how global and regional climate respond to external forcings (Hewitt et al., 2001; Braconnot et al., 2007; Braconnot et al., 2011; Harrison et al., 2014). Previous studies have investigated how the external forcing and boundary conditions during the LGM affected the Intertropical Convergence Zone (ITCZ) (Broccoli et al., 2006; Donohoe et al., 2013; McGee et al., 2014), the Walker circulation (DiNezio et al., 2011), the Indo-Pacific climate (Xu et al., 2010; DiNezio and Tierney, 2013; DiNezio et al., 2016), the SH circulation (Rojas, 2013), and the global monsoon (Jiang et al., 2015; Yan et al., 2016). The Australian monsoon onset and variability during the post-glacial, the late deglaciation, and the Holocene have also been studied using proxy datasets (Ayliffe et al., 2013; De Deckker et al., 2014; Kuhnt et al., 2015; Bayon et al., 2017). However, due to the limitation of the scarce proxy datasets, the Australian monsoon change during the LGM is far from clearly understood.

There are different proxy evidences indicating different Australian monsoon change during the LGM. Here, the Australian monsoon intensity is represented by the seasonality of precipitation, i.e., a stronger monsoon means a wetter summer and/or drier winter. Some records show wet conditions over Australia during the LGM (Ayliffe et al., 2013), while other proxies indicate drier conditions (Denniston et al., 2013; Denniston et al., 2017; DiNezio and Tierney, 2013). The isotopes from eggshell of five regions across Australia affirms that Australia becomes drier in the LGM (Miller et al., 2016), while the archaeological record showed a refugia-type hunter-gatherer response over northwest and northeast Australia during the LGM (Williams et al., 2013), indicating that these areas may have had a wetter summer and were therefore

preferred by people as refugia. The synthesis by the OZ-INTIMATE (Australian INTIMATE,
INTegration of Ice core, MArine and TErrestrial records) project (Turney et al., 2006; Petherick
et al., 2013) showed that the palaeoenvironment over Northern Australia during the LGM was
characterized by drier conditions although wet periods were also noted in the fluvial records
(Reeves et al., 2013a; Reeves et al., 2013b).
The change in the Australian monsoon was inconclusive during the LGM based on proxy
data. Therefore, scholars started investigating the Australian monsoon change from numerical
simulation perspectives. The sensitivity of Australian monsoon to forcings during the late
Quaternary has been analyzed using simulations by Fast Ocean Atmosphere Model (Marshall
and Lynch, 2006, 2008). Numerical experiments have been conducted to analyze the impacts of
obliquity and precession with a coupled General Circulation Model (Wyrwoll et al., 2007) and
orbital time-scale circulation with Community Climate Model (Wyrwoll et al., 2012) on the
Australian monsoon. However, different models may have different responses to the same
external forcings, such that the simulated results may have model dependence. Multi-model
ensemble (MME) can reduce the model biases and therefore provide more reasonable results of
how and why climate system responds to the external forcing changes. The MME can also
provide a clearer perspective on model uncertainties.
Yan et al (2016) thus used the multi-model ensemble approach to examine the response
of global monsoon to the LGM conditions. It was found that the global monsoon and most sub-
monsoons weakened under the LGM conditions. Some brief hypothesis was made to explain the
changes from global and hemispheric perspectives. The Australian monsoon was thought to be
strengthened due to the southward shift of the ITCZ resulted from the hemispheric thermal
contrast and due to the land-sea thermal contrast resulted from the land-configuration. However,
this simulated result of strengthened monsoon or wet condition has not been proved yet. As
mentioned above, it is inconclusive whether the Australian monsoon is strengthened or not
during the LGM, due to the limitations of proxies' and models' uncertainties. Neither model
outputs nor proxy records provide a "true" record of the LGM, as proxy records require
interpretation and calibration and may be spatially incomplete, while models contain biases.
Therefore, model-data and inter-model comparison are needed and studies on the mechanisms
are required to better understand the Australian monsoon change during the LGM. Moreover,
some studies show that the Australian climate during the last glacial period was modulated by
additional mechanisms rather than simply the ITCZ(Bayon et al., 2017). Thus, single forcing
runs are also required to figure out the contributions of different forcings.
This paper investigates the Australian monsoon change during the LGM and its
mechanisms from both thermodynamics and dynamics perspectives, using the multi-model
ensemble mean derived from models in the fifth phase of the Coupled Model Intercomparison
Project (CMIP5) (Taylor et al., 2012) and the third phase of the Paleoclimate Modeling
Intercomparison Project (PMIP3) (Braconnot et al., 2012). We are also trying to quantify the
contributions of the thermodynamic and the dynamic processes to the Australian monsoon
change during the LGM. Additionally, we are applying single forcing run to test the effect of
land-configuration as mentioned in Yan et al. (2016). The models and experiments used in this
paper are introduced in Sect. 2. Section 3 describes simulated results and the physical
mechanisms. The comparison with proxies and other simulations is discussed in Sect. 4 and the
conclusions are made in Sect. 5.

**2 Methods**
2.1 CMIP5/PMIP3 models and experiments
Two experiments performed by models participating in CMIP5/PMIP3 are compared in
this paper: the Last Glacial Maximum Experiment (LGME) and the pre-industrial (PI) control
run (piControl). The models and experiments are listed in Table 1, including 7 models and 2
experiments.
The last 100 years of the LGME and the last 500 years of the piControl from each model
are used to illustrate the model climatology. To obtain the multi-model ensemble (MME), the
model outputs were interpolated into a fixed $2.5\,°$(latitude) $\times 2.5\,°$(longitude) grid using the
bilinear interpolation method.
The LGM external forcing and boundary conditions are listed in Table 2. More specific
documentation can be found on the PMIP3 website (https://pmip3.lsce.ipsl.fr). Compared with
the PI, during the LGM the Southern Hemisphere (SH) low latitudes (30 °S-EQ) receive more
insolation from January to August and less from August to December. The NH low latitudes
(EQ-30 °N) receive less insolation from June to October and more from November to May (Fig.
S1). Due to the decreased sea level, the landmasses expanded during the LGM. A land bridge
formed between Indochina and western Indonesia, and the Arafura Sea between New Guinea and
Australia closed and became landmass (Fig. S2).
To illustrate the robust changes simulated by the different models, the signal-to-noise
ratio (S2N) test is used. The S2N is defined by the ratio of the absolute mean of the MME (as the
signal) to the averaged absolute deviation of the individual model against the MME (as the
noise) (Yan et al., 2016). In the following sections, we only consider the areas in which the S2N
ratio exceeds one when we examine the differences between the LGME and piControl derived
from the MME.
The models contributed to CMIP5 have been evaluated in the previous studies to have
better performance than those in the CMIP Phase 3 (CMIP3) in simulating the Australian
monsoon precipitation seasonality or seasonal cycle (Jourdain et al., 2013; Brown et al., 2016),
which is used to represent the Australian monsoon intensity in this study. However, we need to
keep in mind that there are large uncertainties in model simulations, which require careful
model-data comparison and inter-model comparison.
2.2 NESM model and experiments
To isolate the impacts of land-sea configuration change, two additional sensitivity
experiments are conducted using a newly developed fully coupled earth system model, the
Nanjing University of Information Science and Technology Earth System model version 1
(NESM v1, Cao et al., 2015). One is the piControl run (NESM_PI), which is designed the same
as the PMIP3 protocol. The other is the land sea configuration sensitivity run (NESM_LSM),
which is designed the same as the NESM_PI but with the LGM land sea configuration. The two
experiments are run 500 years after spin-up, and the last 100 years are used.
2.3 Decomposition method
For attribution of precipitation changes, we use a simplified relation based on the
linearized equation of moisture budget used in the previous works (Chou et al., 2009; Seager et
al., 2010; Huang et al., 2013; Endo and Kitoh, 2014; Liu et al., 2016). Considering a quasi-
equilibrium state, the vertical integrated moisture conservation can be written as:

$$-\int_{1000}^{0} \nabla \cdot (q\vec{v})dp = P - E \quad (1)$$

where q is specific humidity, $\vec{v}$ is horizontal velocity, p is pressure, P is precipitation, and E is
the surface evaporation. Since water vapor is concentrated in the lower troposphere, the vertical
integrated total column moisture divergence can be approximately replaced by the integration
from the surface to 500 hPa. Define the $\Delta$ (.) as the change from PI to the LGM, i.e.,
$$\Delta(.) = (.)LGM - (.)PI \qquad (2)$$
Then the precipitation change $\Delta P$ can be calculated as follows:
$$\Delta P = - \int_{p1000}^{p500} \Delta(q \cdot \nabla\vec{v})dp - \int_{p1000}^{p500} \Delta(\vec{v} \cdot \nabla q)\,dp + \Delta E \qquad (3)$$
To further simplify the equation, we use $-\omega_{500}$ to represent vertical integrated $\nabla\vec{v}$, and q at the
surface to represent vertical integrated specific humidity (Huang et al., 2013). Thus, the
precipitation change ($\Delta P$) can be represented as
$$\Delta P \propto \overline{\omega}_{500} \cdot \Delta q + \overline{q} \cdot \Delta\omega_{500} + \Delta E - \Delta T_{adv} \qquad (4)$$
where $\overline{\omega}_{500}$ is 500 hPa vertical velocity in PI, $\overline{q}$ is surface specific humidity in PI, $\Delta T_{adv}$ is the
changes due to the moisture advection ($\int_{p0}^{p500} \Delta(\vec{v} \cdot \nabla q)\,dp$).

The first term in the right-hand side of (4) ($\overline{\omega}_{500} \cdot \Delta q$) represents thermodynamic effect

(due to the change of q), and the second term ($\overline{q} \cdot \Delta\omega_{500}$) represents dynamic effect (due to the
change of circulation).
2.4 Monsoon domain

The monsoon domain is defined following the hydroclimate definition, i.e., a contrast

between wet summer and dry winter (Wang and Ding 2008). The monsoon domain is defined by
the area where the annual range (local summer minus local winter) exceeds 2.0 mm/day, and the
local summer precipitation exceeds 55% of the annual total precipitation. Here in the SH,
summer means November to March and winter means May to September. Since the domains
derived from different models are different, and the changes of domain are also different, we use
the fixed domain derived from the merged Climate Prediction Center Analysis of Precipitation
(CMAP, Xie and Arkin, 1997) and Global Precipitation Climatology Project (GPCP, Huffman et
al., 2009) data.

Note that the monsoon domain is shown to give a general view of precipitation change,

but not the purpose of this study, i.e. the following analysis are not based on the domain.

## 3 Results

We define the difference of precipitation rate between austral summer (DJF) and austral winter (JJA) as the annual range, i.e. the seasonality, to measure the monsoonality of the Australian monsoon. An increased annual range (or seasonality) means a strengthened monsoonality. Unlike the South African and South American monsoon regions (not shown), the monsoonality of the Australian monsoon derived from the seven models' multi-model ensemble (7MME) is strengthened during the LGM (Fig. 1a). This amplified annual range is the result of increased precipitation in austral summer and decreased precipitation in austral winter (Fig. 1b). Note that the largest decrease in precipitation occurs from April to July (late autumn to early winter), not exactly in austral winter; and the largest increase of precipitation occurs in November and December (ND), i.e., austral early summer. Since the amount of autumn–winter reduction of precipitation exceeds the increased precipitation in early summer, the annual mean precipitation over the strengthened annual range region decreases (by 0.36 mm/day). In summary, while the total annual precipitation decreases in the LGM, the annual range (or the seasonality) of the Australian monsoon rainfall is amplified due to seasonal redistribution of the precipitation, especially the drying in austral autumn (April–May) and winter (JJA) over Australia.

Although there are model biases, most of the models (except MPI-ESM-P) simulate an enhanced annual range (or seasonality/monsoonality) in the central Australian monsoon region (20 °S-5 °S, 120 °E-145 °E) (Table 3 and Fig. 1c). Most of the models (except CNRM-CM5 and MPI-ESM-P) also simulate an increased summer precipitation over that region. All the models simulate decreased precipitation from April to September (Fig. 1c). On the other hand, the simulated annual mean precipitation is decreased in most models, except GISS-E2-R. The model uncertainties will be discussed later in Sec. 4.

3.1 Reasons for the decreased precipitation during the LGM in austral winter (JJA)

During the LGM, the lower GHG concentration and the large ice-sheets are the primary causes for the decreased global temperature and humidity. The global surface specific humidity is reduced by 2 g/kg (or 20 %) in JJA during the LGM, compared with the PI. For the SH monsoon regions, the surface specific humidity is reduced more over the Australian monsoon region than over the other two monsoon regions of South Africa and South America (Fig. 2).

As suggested by the Clausius–Clapeyron relation (C-C relation), one degree of
temperature decrease would lead to about a 7 % decrease in the saturation water vapor (Held and
Soden, 2006), or roughly the same scale of decrease in the low tropospheric specific humidity. If
the circulation, evaporation and advection remains unchanged, the precipitation should also be
reduced by 7 % with regard to the equation (4). During the LGM, the simulated near surface air
temperature over the central Australian monsoon region (20 °S-5 °S, 120 °E-145 °E) decreases
significantly by 2.5 K in JJA, which implies a decrease of about 17 % resulted from the C-C
relation. However, the simulated precipitation in the LGM is reduced by 1.45 mm/day or 44 %
comparing to the PI, which is far beyond the value suggested by the thermodynamic effect
(approximately 17 %). This suggests that the majority of the reduction in winter precipitation is
due to the changes of the rest terms of equation (4), including the circulation change (dynamics),
the evaporation change and the change due to the advection term. The changes of each terms
show that the circulation change plays a dominant role in the precipitation change over Australia
(Fig. S3). The change due to the evaporation is also important. The change due to the advection
term is negligible.
The change of the surface wind field shows a strengthened divergence pattern over the
Australian monsoon region (Fig. 3a, vector), which is consistent with the strengthened
descending flow over the Australian monsoon region (Fig. 4) and thus the reduced precipitation
(Fig. 3a, shading). The JJA mean near surface air temperature shows that the land is cooler than
the adjacent ocean around northern Australia (Fig. 5a), which illustrates a strengthened land–sea
thermal contrast because the land cools more than the ocean surface during the LGM. This
strengthened land–sea contrast leads to a higher sea-level pressure (SLP) over land and lower
SLP over ocean in general (Fig. 5b, shading), and thus the outflows from land (Fig. 5b, vector).
The geopotential height at 850 hPa also shows the relative pattern that matches the wind change
(Fig. S4a). The difference of divergence/convergence field (Fig. 5c) also indicates that the
divergence at 850 hPa over northern Australia is strengthened during the LGM. The vertical
velocity at 500 hPa over the central Australian monsoon region (20 °S-5 °S, 120 °E-145 °E)
illustrates that the descending flow strengthened by about 48 %.
In conclusion, both the dynamic process (increased subsidence) and the thermodynamic
process (reduced water vapor content) contribute to the drier winter in the Australian monsoon
region, but the local dynamic processes play a dominant role in the reduction of Australian
winter precipitation.
3.2 Why the precipitation increased in austral early summer (ND)

During ND, the LGM minus PI surface wind difference field shows a strengthened

convergence pattern over the central northern Australian monsoon region (Fig. 6a, vector), which
is consistent with the increased precipitation (Fig. 6a, shading). The vertical velocity at 500 hPa
also shows a strengthened ascending flow over this area (Fig. 7). The increased precipitation
over the central Australian monsoon region is clearly against the thermodynamic effects of the
low GHG concentration and the presence of the ice-sheets, which tends to reduce the
precipitation. The 2-m air temperature is decreased by 2.2 K and the surface specific humidity is
reduced by 2.6 g/kg (or 16.0 %) over the Australian monsoon region (Fig. 8). The precipitation
would decrease by 15.4 % according to the thermodynamic effect without the circulation change.
However, the precipitation over the Australian monsoon region is increased by about 13.0 %.
Therefore, the changes in dynamic processes must induce a 29 % increase of precipitation, so
that the net increase in precipitation reaches 13 %.

There is a cyclonic wind anomaly associated with an anomalous low pressure over the

northwest Australia (Fig. 6a and Fig. 9b, vector), accompanied by a strengthened low-level
convergence (Fig. 9c), which favors increased precipitation in the Australian monsoon region.
The change of the moisture transport (moisture flux) also indicates increased moisture transport
into northern Australia (not shown). The cyclonic vorticity in northwest Australia is partially
caused by the enhanced strong low-level westerlies that prevail north of Australia.

We now seek to determine why there is a strengthened low-level westerly with maximum

over north of Australia. We first consider the temperature change. The ND mean 2-m air
temperature during the LGM shows that the two enlarged landmasses over the Indo-Pacific
warm pool region (resulting from the lower sea level) change differently (Fig. 9a). It is cooler
over the northwest landmass (western Indonesia–Indochina) and relatively warmer over the
southeast landmass (eastern Indonesia–northern Australia). This temperature variation forms a
southeast–northwest temperature gradient (Fig. 9a, Fig. S5a, S5b), accompanied by a northwest–
southeast SLP gradient (Fig. 9b, Fig. S5c, S5d). The northwest–southeast pressure gradient is
stronger in the geopotential height change at 850 hPa (Fig. S4b). The high pressure in the
western Indonesia–Indochina is a part of the larger scale enhanced winter monsoon over the
South China Sea. This enhanced winter monsoon flows cross the equator from the NH to the SH
and turn into strong westerlies due to deflection induced by the Coriolis force. The 850 hPa
convergence strengthens over the Australian monsoon region (Fig. 9c), and the corresponding
ascending motion at 500 hPa also increases over the Australian monsoon region.

Another reason for circulation change is the sea surface temperature (SST) gradient

change. The SST anomaly in ND shows a warmer Western Pacific and cooler Eastern Indian
Ocean pattern (Fig. 10), indicating a westward temperature gradient (Fig. S5e), and thus an
eastward pressure gradient which, in the equatorial region, can directly enhance westerly winds
near the northern Australian monsoon region (Fig. 9b). Li et al. (2012) also found that a cold
state of the Wharton Basin (100°E–130°E, 20°S–5°S) was accompanied by anomalous westerlies
and cyclonic circulation anomalies in the Australian monsoon region, which were associated
with a strong tropical Australian summer monsoon and enhanced rainfall over northeast
Australia.

In summary, during ND, the enlarged land area due to sea-level drop enhances the land–

sea thermal contrast, and forms a northwest–southeast thermal contrast which induces low
pressure over northern Australia but high pressure over the adjacent ocean and the Indochina–
western Indonesia, leading to the enhanced convergence over northern Australia and thus
increasing the early summer monsoon rainfall. The SST gradients between the warm equatorial
western Pacific and the relatively cool eastern Indian Ocean during the pre-summer monsoon
season also contribute to the strengthened equatorial westerlies and the cyclonic wind anomaly
over northern Australia. These dynamic mechanisms have a positive contribution to the early
summer precipitation. The thermodynamic effects have negative contribution to the precipitation
change, but with smaller magnitude. Therefore, the early summer precipitation over northern
Australia increases. We can also tell from the changes of the decomposed terms that the
dynamics plays much more important role in the precipitation change over Australia, especially
the distribution pattern (Fig. S6). Again, the impacts of evaporation and advection terms are
small.

**4 Discussion**
The intensification of the Australian monsoon in this study is measured by the enhanced
seasonal difference (or the seasonality) of precipitation, and is particularly attributed to the
decreased austral winter precipitation. This is consistent with the reconstructed results by
Mohtadi et al. (2011), which indicated that it was not significantly drier in austral summer during
the LGM, while the winter monsoon was as weak as the modern period. Whereas the annual
mean precipitation is decreased, which means the Australian monsoon would be weakened
during the LGM when it is measured by the annual mean precipitation. The modeling study by
DiNezio et al. (2013) suggested a decreasing rainfall across northern Australia during the LGM,
consistent with the proxy synthesis by stalagmite (Denniston et al., 2017). The decreased rainfall
in their work represents the annual mean precipitation, which is also consistent with our work.
On the other hand, the increased rainfall in austral summer in this study is consistent with what
has been revealed in the reconstructed work by Liu et al. (2015), which found intense austral
summer precipitation over Papua New Guinea and North Australia in LGM. The decreased
annual mean precipitation and the intensified seasonality of precipitation over the Australian
monsoon region are in agreement with the synthesis from the simulated result by Tharammal et
al. (2013) using a set of experiments.
For the forcings and mechanisms of the Australian monsoon change during the LGM,
there are large changes in four external forcings during the LGM, including the insolation change
resulting from the orbital change, the land–sea configuration change, the GHG change and the
presence of ice-sheets. The lower GHG concentrations and the presence of ice-sheets are likely
to be contributors to the thermodynamic effect leading to the reduced water vapor and thus the
decreased rainfall both in austral winter and early summer. The enlarged the landmasses over
western Indonesia and northeastern Australia are essential to the local dynamic processes that
influence the rainfall. The low obliquity and high precession during the LGM may be another
factor that can affect the rainfall (Liu et al., 2015). However, the impact of the insolation change
caused by the orbital change remains unknown.
During the LGM, the insolation over tropical region increased from December to June
and decreased from July to November relative to the present day (Fig. 11a). In the annual
variation, the precipitation responds to the lower tropospheric moisture convergence. The
moisture change depends on temperature change while the circulation change depends on surface
temperature gradients change. The change of the surface temperature lags insolation changes
because the ocean and land surfaces have heat capacity (thermal inertial). In other words,
insolation is a heating rate which equals to temperature change (tendency) but not the
temperature itself. Thus the precipitation change would lag the insolation change by about two
months, due to the ocean–atmosphere interaction without other processes. However, the
simulated Australian monsoon precipitation is decreased from March to September and increased
from November to February (Fig. 11b), quite different from what it would be (i.e., decrease from
September to January and increase from February to August). This indicates the insolation
change might have little effect on the Australian monsoon precipitation. Meanwhile, the
insolation over SH is increased during the LGM from April to August, when Australia is in late
autumn and winter. An increased insolation might make land warmer than ocean thus against the
climatology (i.e. cooler land and warmer ocean in winter). However, the simulated surface
temperature reduced more over Australia than the adjacent oceans (Fig. 5a). On the other hand,
the synthesis of Wyrwoll et al. (2007) and Liu et al. (2015) indicates the strong convergence rain
belt (ITCZ) stays in the north during those times with low obliquity and high precession, which
is a little more northerly than that stays in our study. These mean that the effect of orbital change
and thus the insolation change might be suppressed by other factors.
Moreover, the paleoclimate records suggest that it was dry and cool in the Indo-Pacific
Warm Pool region during the LGM (Xu et al., 2010). The simulated SST is consistent with the
reconstructions. Although in the early austral summer, over the Indian Ocean warm pool, it is
cooler over the SH, while over the Pacific warm pool, it is cooler over the NH (Fig. S7). Such
anomalous SST asymmetry may favor the southward shift of the ITCZ over Australia and the
southwest Pacific, which might be related to the enhanced austral summer monsoon
precipitation. However, the 7MME shows no significant ITCZ shift during the LGM, particularly
over the central Australian monsoon region (Fig. S8). The reconstructions and simulations by
McGee et al. (2014) and Mohtadi et al. (2014) also suggested that there was no significant shift
of ITCZ position during the LGM.
Therefore, it is the local dynamical processes, instead of the large-scale circulation such
as the position of the ITCZ induced by the NH-SH thermal contrast, that might be the key factor
influencing the early summer mean precipitation change over the Australian monsoon region
during the LGM. To test this hypothesis, we compared the results from the two additional runs,
the NESM_PI and the NESM_LSM. The changes of the ND mean precipitation and wind field at
1000 hPa between the NESM_LSM and the NESM_PI are similar to the changes derived from
the 7MME, i.e. the precipitation is increased and the convergence is strengthened over northern
Australia (Fig. 12a). The changes of the 2-m air temperature, SLP and 850 hPa wind field (Fig.
12b, c) are also similar to those results in the 7MME (Fig. 9). It is also cooler over the northwest
landmass (western Indonesia–Indochina) and relatively warmer over the southeast landmass
(eastern Indonesia–northern Australia) (Fig. 12b). This temperature variation is also
accompanied by a northwest–southeast SLP gradient and the strengthened cross equatorial flow
converging to north Australia (Fig. 12c). This sensitivity simulation confirms that the local
dynamical processes induced by the land sea configuration are essential to the Australian
monsoonality change.

Although the 7MME simulates strengthened Australian monsoonality, there are

uncertainties among individual models. The most notable uncertainty is the increased austral
summer (DJF) precipitation. Five out of the 7 models simulate increased DJF mean precipitation
over the Australian monsoon region during the LGM (CCSM4, GISS-E2-R, IPSL-CM5A-LR,
MIROC-ESM and MRI-CGCM3), while the other two simulate decreased precipitation (CNRM-
CM5 and MPI-ESM-P) (Fig. 13), especially over the land area. The wind field at 850hPa
geopotential height shows a cyclonic anomaly pattern over northern Australia in the five models
(Fig. 14a), accompanied with a strengthened ascending flow (not shown). While in the other two
models, there is no cyclonic wind anomaly over Australia region (Fig. 14b), and the ascending
flow is weakened (not shown). The different changes of wind field indicate the different
precipitation responses to the LGM boundary conditions in the two model groups.

The austral spring and summer mean 2m-air temperature and SST also change differently

in these two model groups. The main differences are located over the tropic Pacific Ocean and
the North Atlantic Ocean. It is cooler over high-latitude Northern Atlantic Ocean in the five
models, whereas warmer in the two models, mainly in the austral spring (Fig. 15a, 15b). In the
austral summer, there is an eastern Pacific El Nino-like patter in the five models, while there is a
Central-Pacific El Nino (CP-El Nino) like pattern in the two models (Fig. 15c, 15d). Studies have
shown that the CP-El Nino is related to the Asian-Australian monsoon system (Yu et al. 2009),
and would lead to a markedly decreased precipitation in December (Taschetto et al. 2009).

Therefore, the different SST responses over the Pacific Ocean and the north Atlantic

Ocean in austral spring and summer in different models might be the key factor that leads to
different wind anomalies and thus different Australian monsoon precipitation changes. Note that
the resolution of land configuration in each model might not be the key factor that affects the
SST gradient over the eastern Indian Ocean and western Pacific Ocean (Fig. S9).

**5 Conclusions**

The global mean temperature and water vapor have an overall decrease under the LGM
forcings (lower GHG and large ice-sheets). Nevertheless, the simulated Australian monsoon
seasonality derived from CMIP5/PMIP3 multi-model ensemble has a distinctive amplification
(or the monsoonality is intensified) against the weakened global monsoons elsewhere during the
LGM. This study then investigated the possible reasons for this strengthened Australian
monsoonality from both a thermodynamic and dynamic perspective.
The conclusions are as follows:
1)  The Australian monsoon seasonality is strengthened as a result of the enhanced

seasonal difference between austral summer and winter, i.e., the increased early

summer (ND) mean rainfall and the reduced winter (JJA) mean rainfall. Both the

dynamic processes and thermodynamic effects contribute to the precipitation change;

however, the dynamic processes have a much stronger contribution than the

thermodynamic effects.

2)  The Australian winter (JJA mean) precipitation derived from 7MME is decreased

during the LGM relative to the preindustrial control experiment. The dynamic

processes, induced by the enhanced land–ocean thermal contrast, contribute more to the

decreased rainfall through the strengthened divergence over northern Australia (Fig.

16a), whereas the thermodynamic effect (i.e., the reduced atmospheric water vapor due

to the lower temperature induced by lower GHGs and present ice-sheets) and

evaporation have moderate contributions.

3)  For the increased precipitation in early summer (ND) in the 7MME, the local dynamic

processes have a positive contribution and the thermodynamic effect has a negative

contribution. Both the decomposition method and the sensitivity simulations show that

the dynamic effect plays most important role for the increased rainfall. The local

dynamic processes are mainly induced by the northwest–southeast thermal contrast
between Indochina–western Indonesia and northeastern Australia. The east Indian
Ocean–west Pacific Ocean thermal gradient also contributes to these processes (Fig.
16b).
4)   The sensitivity simulation illustrates that the change in circulation over Australia is
very likely to be rooted in the enlarged landmasses over the Indochina–western
Indonesia and New Guinea, and northern Australia. Another factor contributes to the
circulation change might be the asymmetric change between western Pacific Ocean and
eastern Indian Ocean. These have critical impacts on the thermal gradients that induce
changes in the low-level circulation pattern and convergence/divergence.
Note that models have uncertainties, i.e. not all the models simulate an intensified
seasonality of Australian monsoon. The different SST responses over Pacific Ocean and Atlantic
Ocean in different models to the same external forcings are essential for the model uncertainties.
More model-data comparison and inter-model comparison are required to better understand the
model-data disagreement and improve confidence in model results.
Our results are based on the equilibrium simulation, representing a mean state of the
Australian monsoon change and its possible mechanisms during the LGM. More simulations
with single forcing (such as the SST asymmetry change, the insolation change) are required to
further understand the effect of each factor and to specifically quantify the contribution of each
forcing to the Australian monsoon change.

**Acknowledgments**

We acknowledge Prof. Williams J and the two reviewers for the comments helping to clarify and improve the paper. This research was jointly supported by the National Key Research and Development Program of China (Grant No. 2016YFA0600401), the National Basic Research Program (Grant No. 2015CB953804), the National Natural Science Foundation of China (Grant Nos. 41671197, 41420104002 and 41501210), and the Priority Academic Development Program of Jiangsu Higher Education Institutions (PAPD, Grant No. 164320H116). We acknowledge the World Climate Research Programme's Working Group on Coupled Modeling, which is responsible for the CMIP, and we thank the climate modeling groups for producing and making available their model outputs. For the CMIP, the U.S. Department of Energy's Program for climate model diagnosis and intercomparison provided coordinating support and led the development of software infrastructure in partnership with the Global Organization for Earth System Science Portals. We thank LetPub (www.letpub.com) for its linguistic assistance during the preparation of this manuscript. This is the ESMC publication XXX.

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

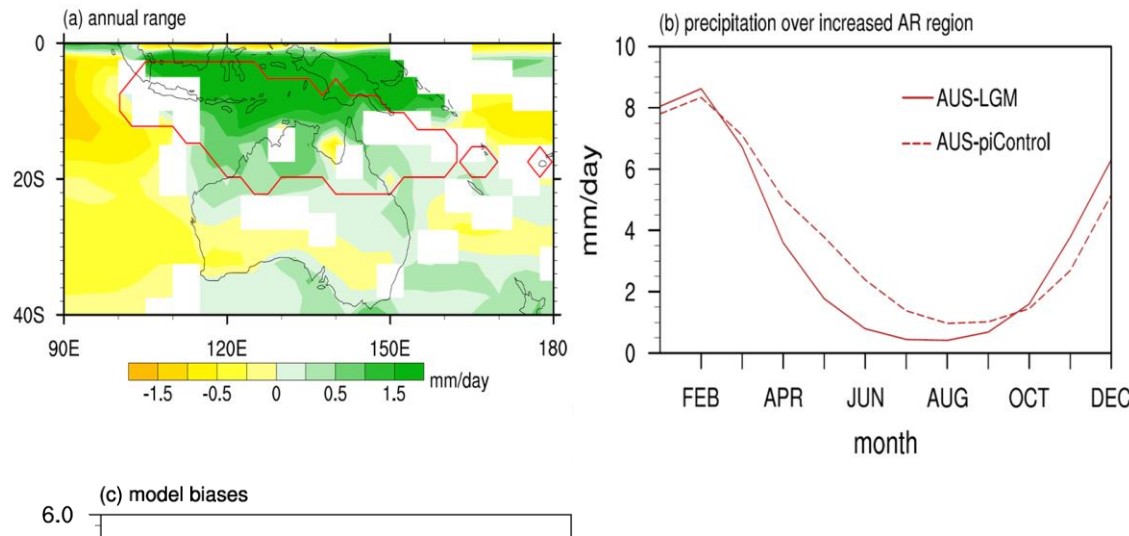

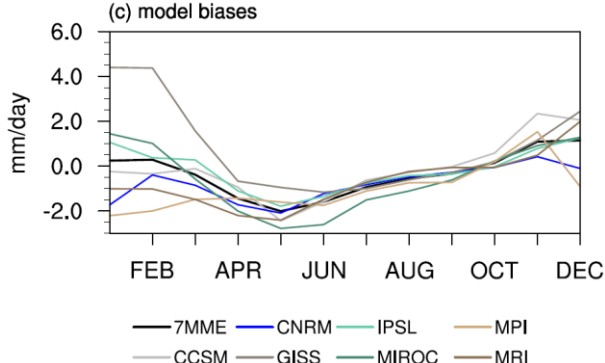

**Figure 1** (**a**) Spatial distribution of changes in the annual range (AR) of precipitation measured by the difference between LGME and piControl, (**b**) seasonal distribution of the precipitation in the increased AR area (20˚S-5˚S, 120˚E-145˚E), and (c) seasonal distribution of the precipitation differences in the increased AR area derived from 7 MME (black line) and each model (colored lines). The red solid line in (**a**) encloses the Australian monsoon rainfall domain. The dashed (solid) line in (**b**) denotes the seasonal distribution of precipitation derived from the piControl (LGME) run. Only those areas where signal-to-noise ratio exceeds one are plotted in (**a**).

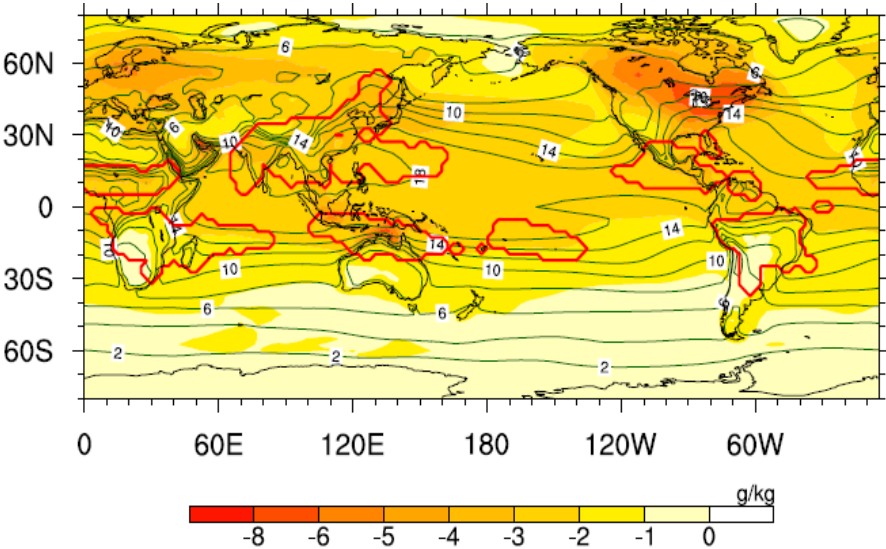

658

**Figure 2** Difference of JJA mean surface specific humidity between LGME and piControl
(shaded). The green contours denote the climatology derived from piControl. The red lines
enclose the monsoon domains. Only those areas where signal-to-noise ratio exceeds one are
plotted.

663

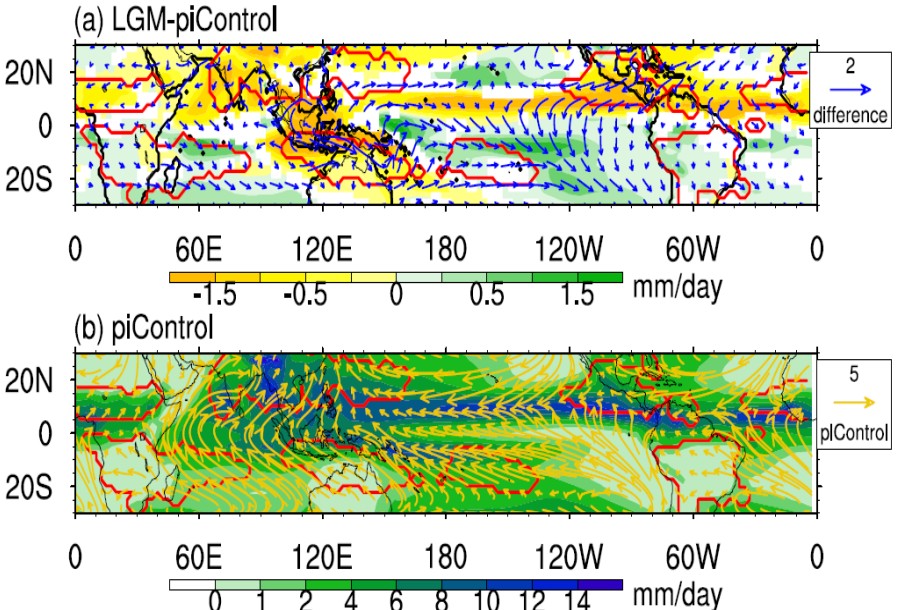

664

**Figure 3** (**a**) JJA mean precipitation (shading) difference and surface wind (vectors) difference
between LGME and piControl, and (**b**) the climatology of JJA mean precipitation (shading) and
surface wind (vectors) derived from piControl. The red lines enclose the monsoon domains. The
thick black lines in (**a**) denote the coastal lines in LGME provided by CMIP5/PMIP3, and the

thin black lines denote the coastal lines in piControl. Only those areas where signal-to-noise ratio
exceeds one are plotted in (**a**).

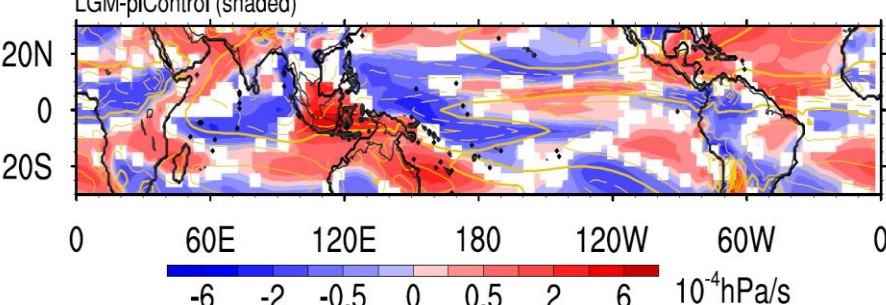


**Figure 4** The difference of JJA mean vertical velocity at 500 hPa between LGME and piControl
(in shading) and the corresponding climatology derived from piControl (yellow contours). The
thick black lines denote the coastal lines in LGME provided by CMIP5/PMIP3, and the thin
black lines denote the coastal lines in piControl. Only those areas where signal-to-noise ratio
exceeds one are plotted in the difference pattern.

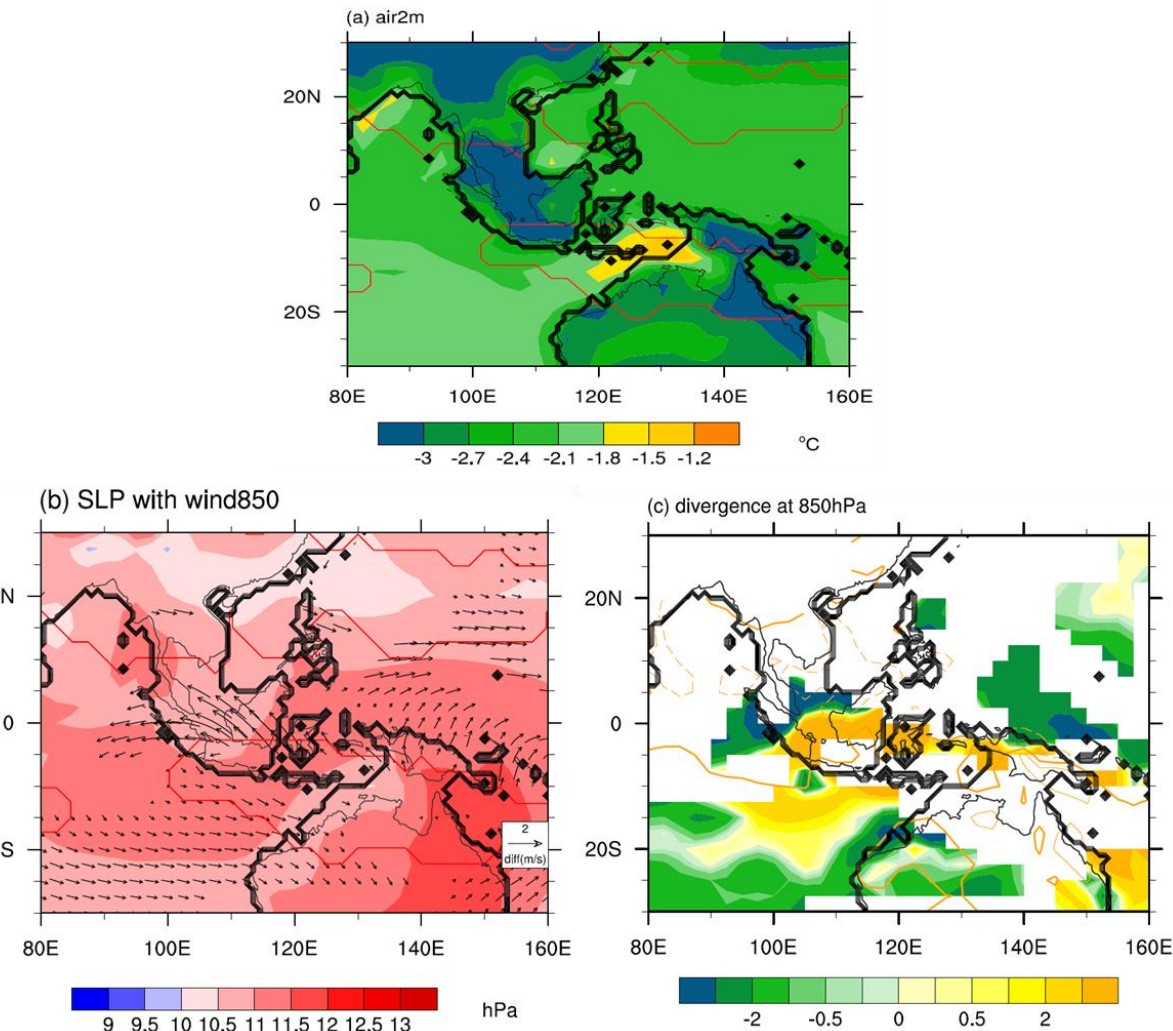

**Figure 5** JJA mean (**a**) surface air temperature, (**b**) sea level pressure (shading) with 850 hPa
wind (vector), and (**c**) 850 hPa divergence differences between LGME and piControl. The red
lines in (**a**) and (**b**) enclose the monsoon domains. The orange lines in (**c**) represent the
climatology derived from piControl. The thick black lines denote the coastal lines in LGME
provided by CMIP5/PMIP3, and the thin black lines denote the coastal lines in piControl. Only
those areas where signal-to-noise ratio exceeds one are plotted.

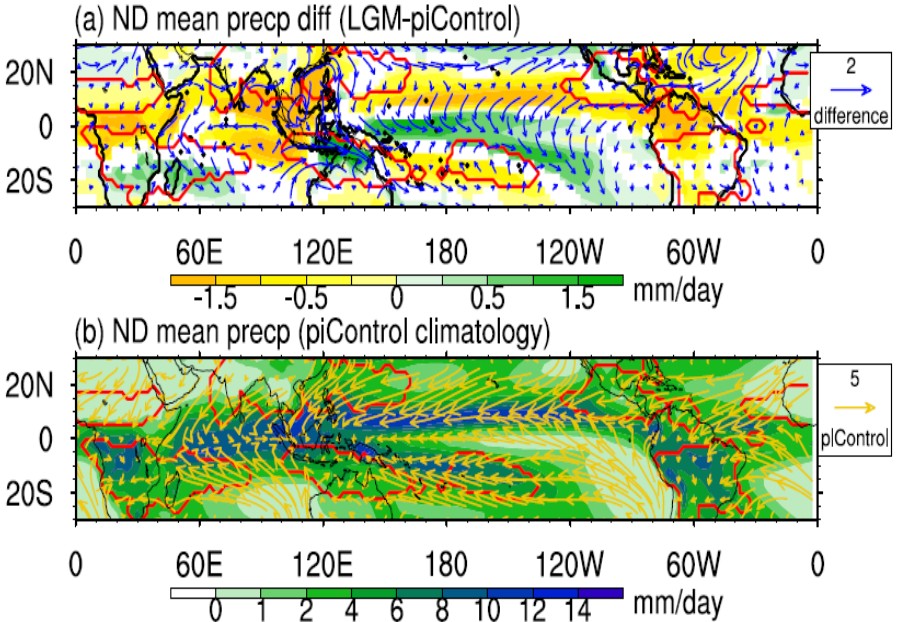


**Figure 6** (**a**) ND mean precipitation (shading) difference and surface wind (vectors) difference between LGME and piControl, and (**b**) the climatology of ND mean precipitation (shading) and surface wind (vectors) derived from piControl. The red lines enclose the monsoon domains. The thick black lines in (**a**) denote the coastal lines in LGME provided by CMIP5/PMIP3, and the thin black lines denote the coastal lines in piControl. Only those areas where signal-to-noise ratio exceeds one are plotted in (**a**).

694

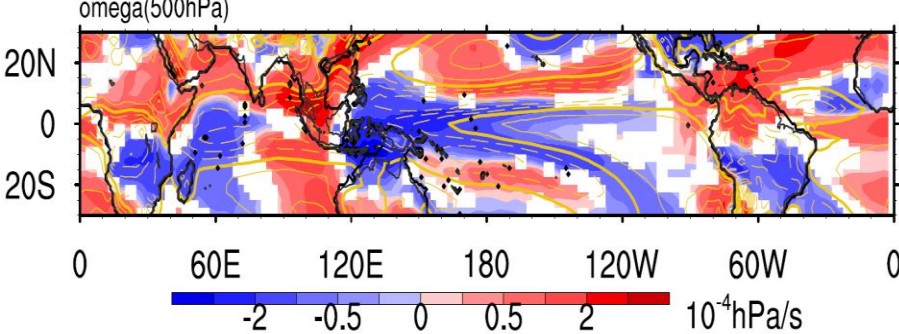

695

**Figure 7** The difference of the ND mean vertical velocity at 500 hPa between LGME and piControl (in shading) and the corresponding climatology derived from piControl (yellow contours). The thick black lines denote the coastal lines in LGME provided by CMIP5/PMIP3, and the thin black lines denote the coastal lines in piControl. Only those areas where signal-to-noise ratio exceeds one are plotted in the difference pattern.


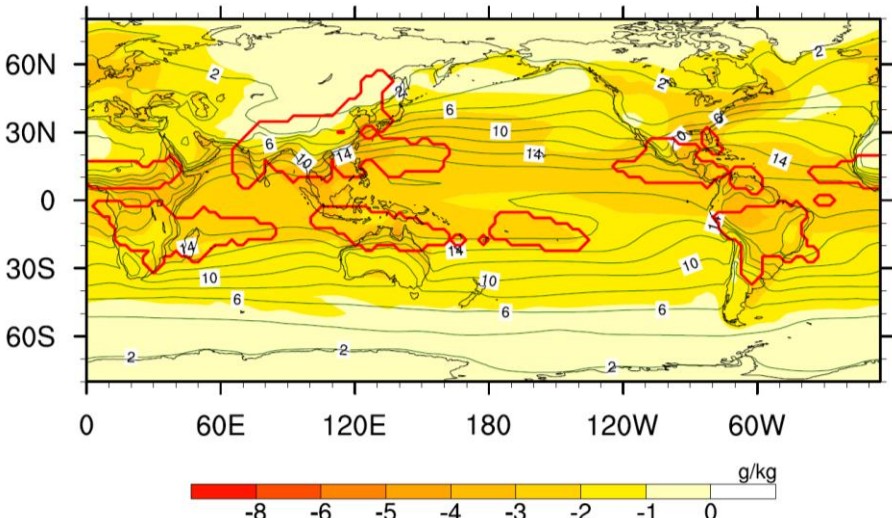

**Figure 8** Difference of ND mean surface specific humidity between LGME and piControl
(shaded). The green contours denote the climatology derived from piControl. The red lines
enclose the monsoon domains. Only those areas where signal-to-noise ratio exceeds one are
plotted.

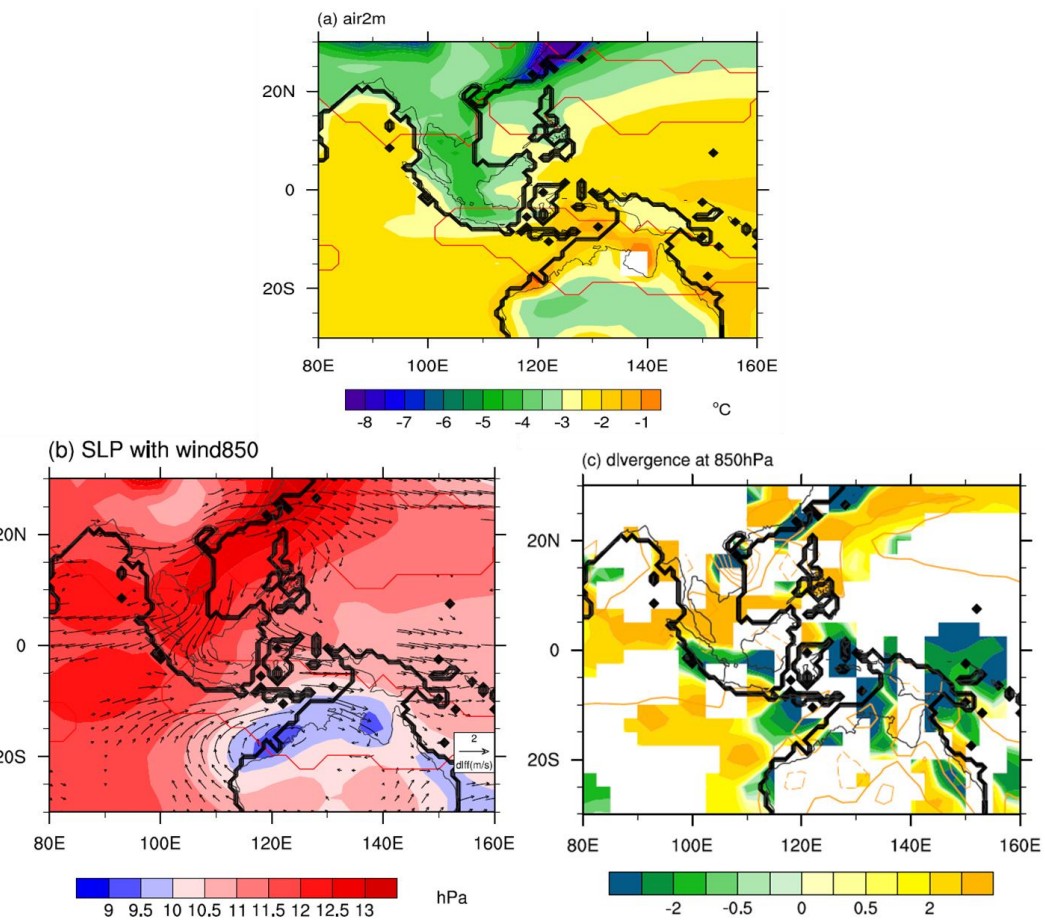

**Figure 9** ND mean (**a**) surface air temperature, (**b**) sea level pressure (shading) with 850 hPa

wind (vector), and (**c**) 850 hPa divergence difference between LGME and piControl (shading).

The red lines in (**a**) and (**b**) enclose the monsoon domains. The orange lines in (**c**) represents the

climatology derived from piControl. The thick black lines denote the coastal lines in LGME

provided by CMIP5/PMIP3, and the thin black lines denote the coastal lines in piControl. Only

those areas where signal-to-noise ratio exceeds one are plotted.

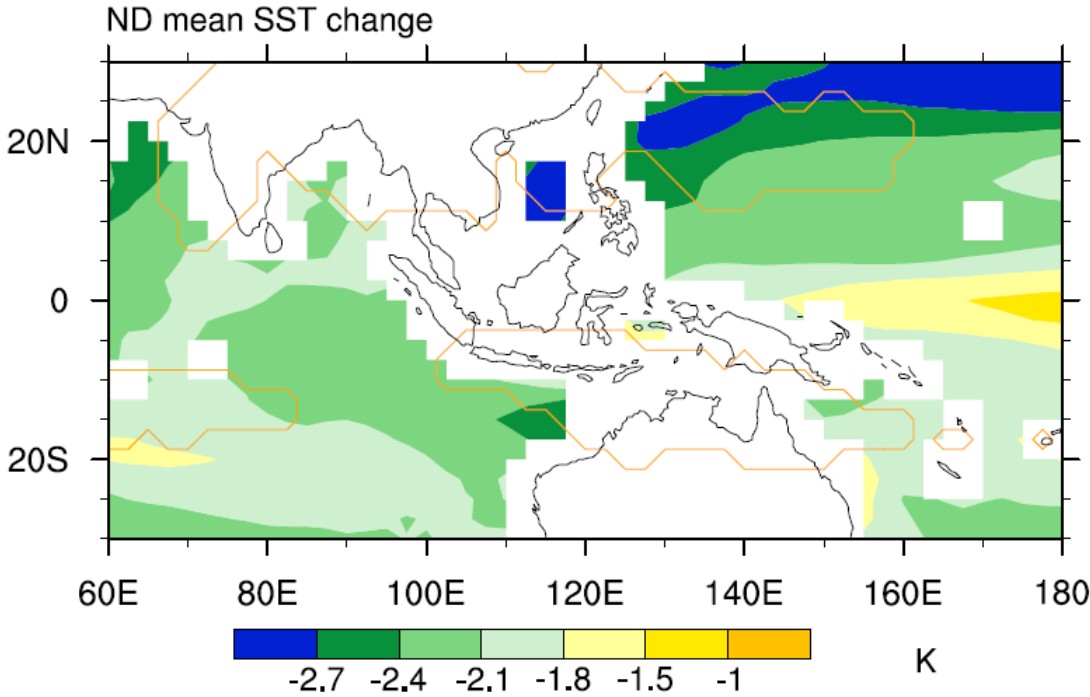


**Figure 10** ND mean SST difference between LGME and piControl. The red lines enclose the
monsoon domains. Only those areas where signal-to-noise ratio exceeds one are plotted.

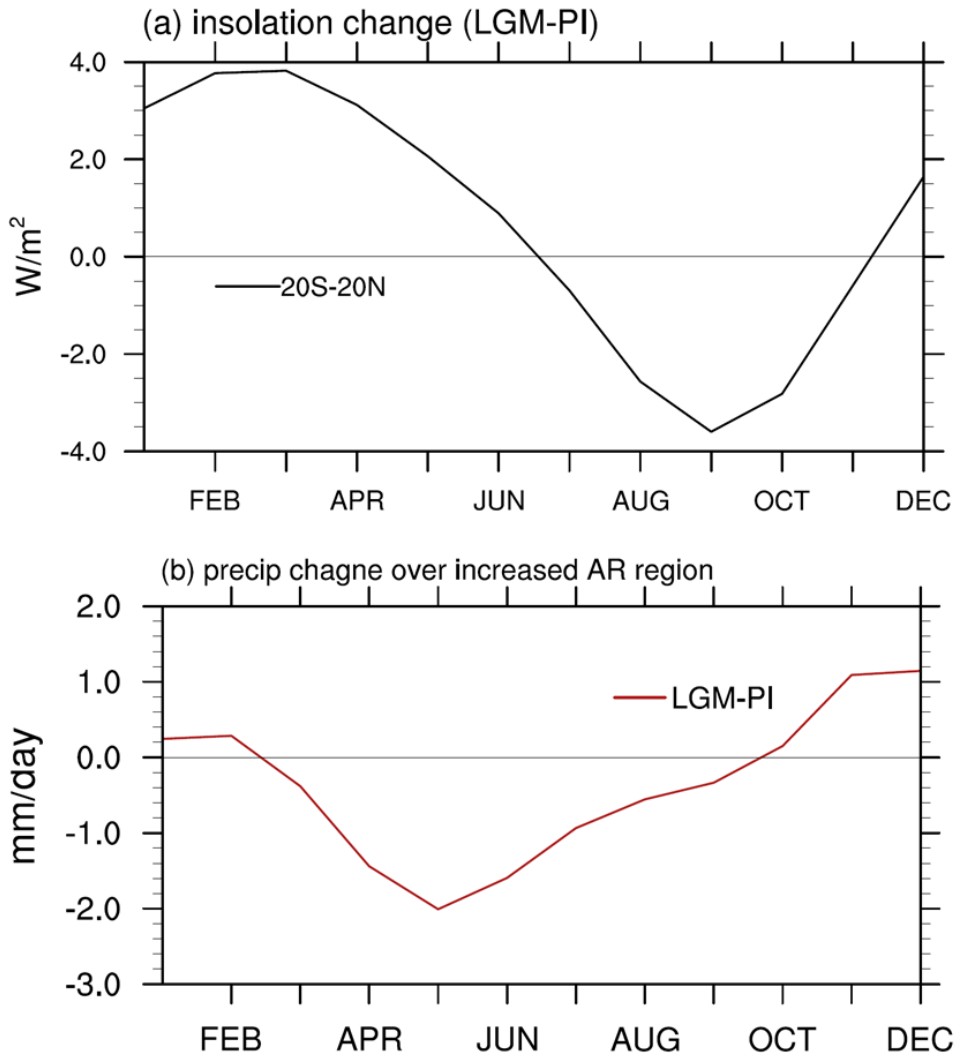


**Figure 11** Seasonal distribution of (**a**) insolation change between 20 ˚S and 20 ˚N, and (**b**)

precipitation change over the increased AR region as indicated in Fig. 1b (20 ˚S-5 ˚S, 120 ˚E-

145 ˚E). The changes are calculated by the LGM value minus the PI value.








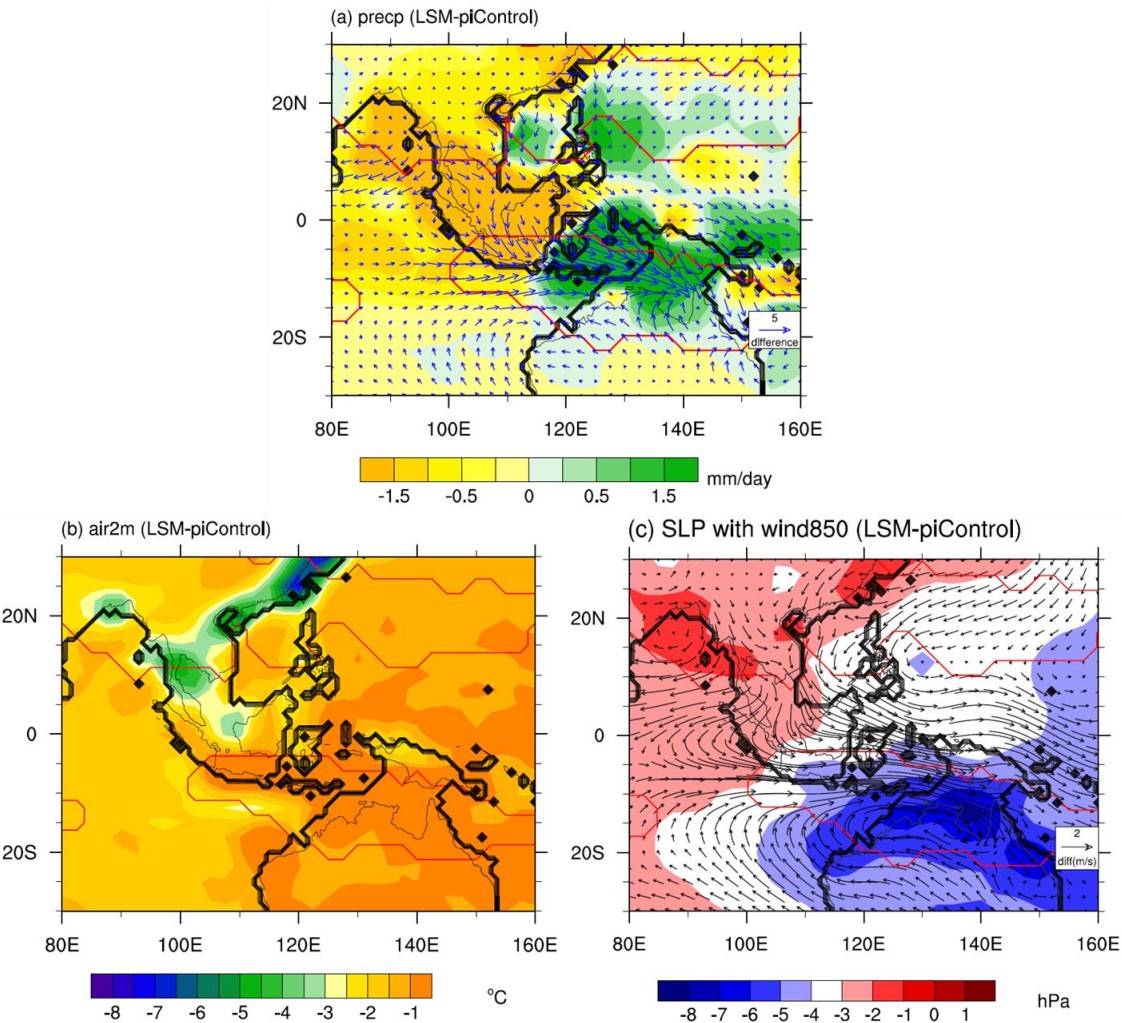


**Figure 12** ND mean (a) precipitation (shading) with 1000 hPa wind (vector), (b) surface air

temperature, and (c) sea level pressure (shading) with 850 hPa wind (vector) difference between

the NESM_LSM and the NESM_PI. The red lines enclose the monsoon domains. The thick

black lines denote the coastal lines in NESM_LSM, and the thin black lines denote the coastal

lines in NESM_PI.

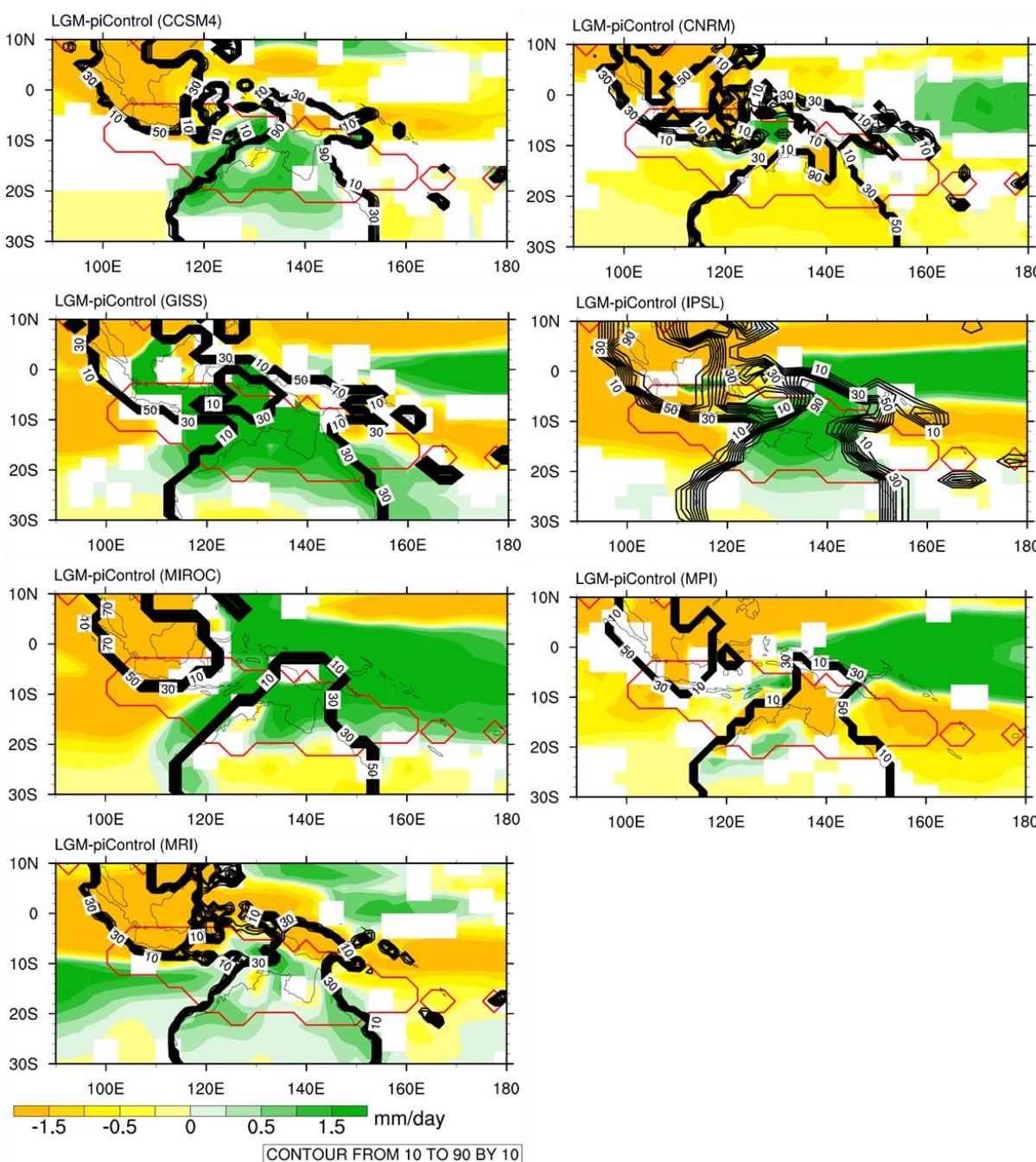

**Figure 13** DJF mean precipitation differences between LGME and piControl derived from each model. The red lines enclose the monsoon domains. The dark black lines show the land area fraction used for the LGME in each model. Only those areas where signal-to-noise ratio exceeds one are plotted.



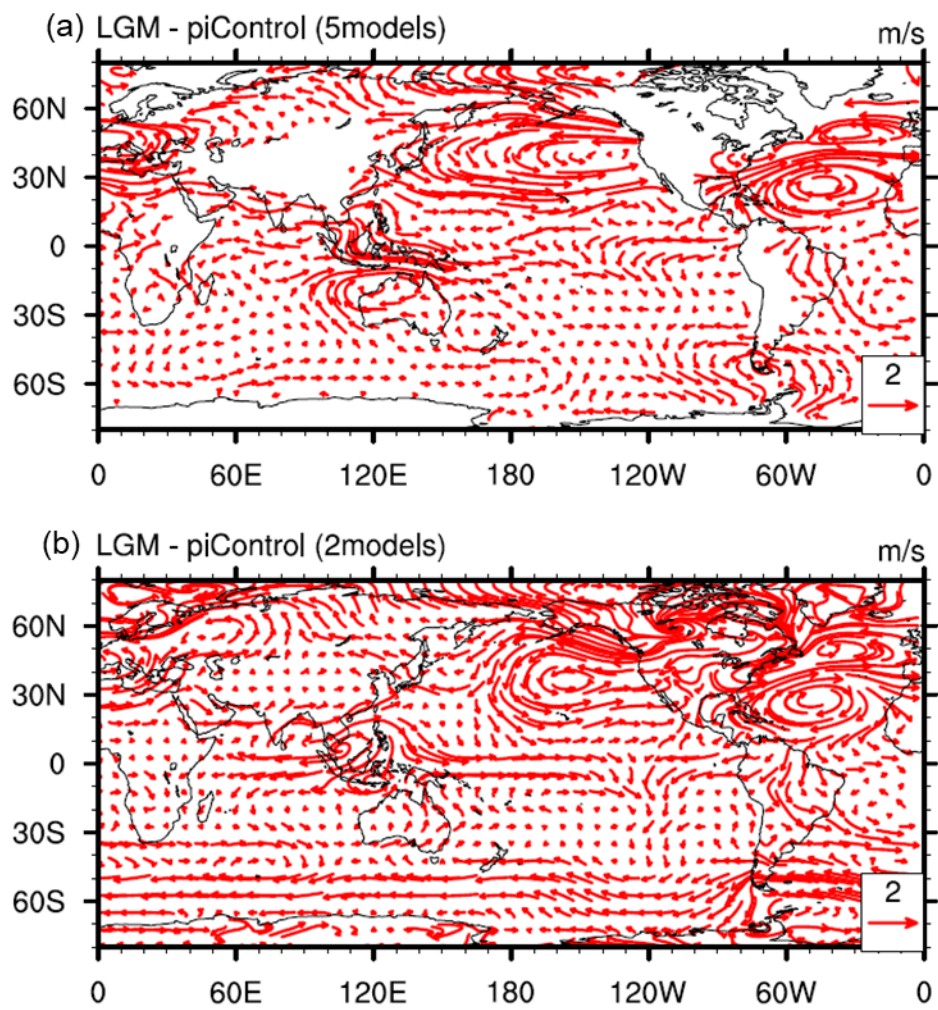


**Figure 14** DJF mean 850hPa wind differences between LGME and piControl derived from (a) the five models and (b) the two models. Only those areas where signal-to-noise ratio exceeds one are plotted.


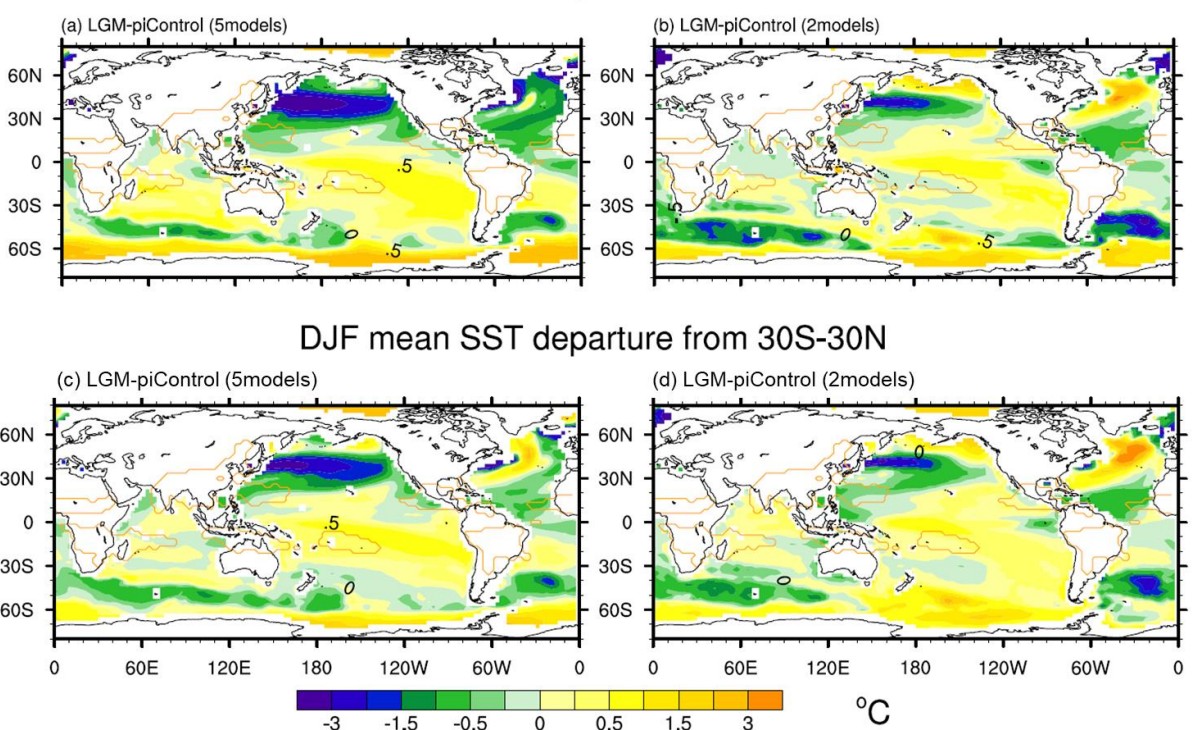

**Figure 15** SON mean (a)-(b) and DJF mean (c)-(d) SST differences between LGME and piControl derived from (a), (c) the five models and (b), (d) the two models. Only those areas where signal-to-noise ratio exceeds one are plotted. The area average of tropical (30 °S-30 °N) SST change is distracted to make it clearer to illustrate the regional differences.

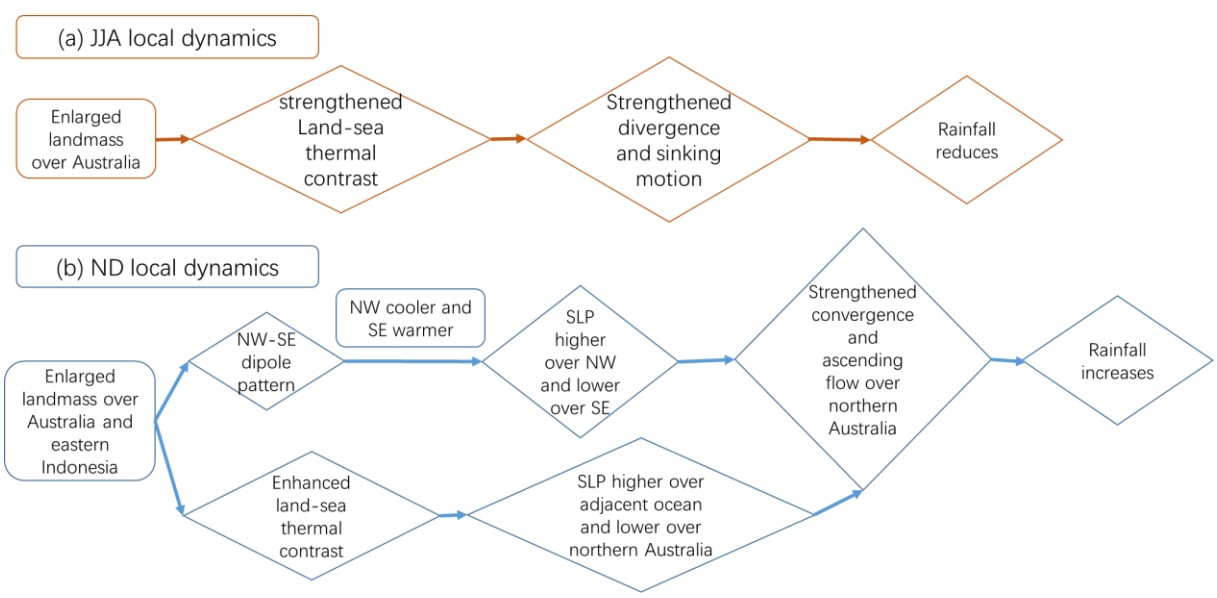

**Figure 12** Mechanisms of Australian monsoon precipitation change (**a**) in JJA, and (**b**) in ND
during the LGM, in  the local dynamics perspective.



**Table 1** CMIP5/PMIP3 models and experiments used in this study.

| Model | Institution | piControl Time span (years) | LGME Time span (years) | Spatial resolution for atmospheric module Lon × Lat Grids | Spatial resolution for oceanic module Lon × Lat Grids |
|---|---|---|---|---|---|
| **CCSM4** | National Centre for Atmospheric Research (NCAR) | 501 | 101 | 288 × 192 | 320×384 |
| **CNRM-CM5** | Centre National de Recherches Meteorologiques/Centre Europeen de Recherche et Formation Avancees en Calcul Scientifique (CNRM-CERFACS) | 850 | 200 | 256 × 128 | 362×292 |
| **GISS-E2-R** | NASA Goddard Institute for Space Studies (NASA GISS) | 1200 | 100 | 144 × 90 | 288×180 |
| **IPSL-CM5A-LR** | Institute Pierre-Simon Laplace (IPSL) | 1000 | 200 | 96 × 95 | 182×149 |
| **MIROC-ESM** | Atmosphere and Ocean Research Institute, University of Tokyo, National Institute for Environmental studies, and Japan Agency for Marine-Earth Science and Technology | 531 | 100 | 128 × 64 | 256×192 |
| **MPI-ESM-P** | Max Planck Institute for Meteorology | 1156 | 100 | 196 × 98 | 256×220 |
| **MRI-CGCM3** | Meteorological Research Institute (MRI) | 500 | 100 | 320 × 160 | 364×368 |

**Table 2** Main changed boundary conditions used for the piControl and LGME experiments.

| | piControl | LGME |
|---|---|---|
| **Orbital parameters** | Eccentricity = 0.016724<br>Obliquity = 23.446°<br>Angular precession = 102.04° | Eccentricity = 0.018994<br>Obliquity = 22.949°<br>Angular precession = 114.42° |
| **Trace gases** | $CO_2$ = 280 ppm<br>$CH_4$ = 650 ppb<br>$N_2O$ = 270 ppb | $CO_2$ = 185 ppm<br>$CH_4$ = 350 ppb<br>$N_2O$ = 200 ppb |
| **Ice sheets** | Modern | Provided by ICE-6G v2 (Peltier, 2009) |

| | | | |
|---|---|---|---|
| **Land surface elevation and coastlines** | Modern | | Provided by PMIP3 |


Table 3 Annual mean, austral summer (DJF) mean and annual range of precipitation change over the region of (20 °S-5 °S, 120 °E-145 °E). The area averaged value is calculated based on the areas where S2N ratio exceed one.

| Model | Annual mean (mm/day) | Summer mean (mm/day) | Annual range (mm/day) |
|---|---|---|---|
| CCSM4 | -0.14 | 0.49 | 1.36 |
| CNRM-CM5 | -0.78 | -0.74 | 0.12 |
| GISS-E2-R | 0.79 | 3.74 | 4.66 |
| IPSL-CM5A-LR | -0.17 | 0.90 | 1.82 |
| MIROC-ESM | -0.53 | 1.25 | 3.17 |
| MPI-ESM-P | -1.02 | -1.71 | -0.52 |
| MRI-CGCM3 | -0.68 | -0.01 | 0.85 |
| 7MME | -0.36 | 0.56 | 1.61 |
