# Peer review of "Understanding the Australian Monsoon change during the Last Glacial Maximum with multi-model ensemble"

_Climate of the Past, 2018_

## Short Comment (SC1) · 18 Apr 2018

Dear authors, I am surprised that results from several modeling and proxy studies on the LGM climate over the Australian-Indonesian monsoon region are missing. Your inference that the ITCZ did not shift during the LGM is already shown by Mohtadi et al. (2014) using both CCSM3 model and proxy data. However, that study argues that annual rainfall did not change in the western part of the Australian-Indonesian monsoon system during the LGM (but during the Heinrich stadial). Reference: Mohtadi M., Prange M., Oppo D.W., De Pol-Holz R., Merkel U., Zhang X., Steinke S., Lückge A., 2014. North Atlantic forcing of tropical Indian Ocean climate. Nature 509, 76-80.

[Figure]

Further, both the austral summer and winter monsoon changes since the LGM have been reconstructed by Mohtadi et al. (2011), showing that the austral summer monsoon does not show glacial-interglacial variability, and that the LGM was not significantly drier than the other periods, while the austral winter monsoon was as weak as today during the LGM and follows the Northern Hemisphere summer insolation. Reference: Mohtadi M., Oppo D.W., Steinke S., Stuut J-B W., De Pol-Holz R., Hebbeln D., Lückge A., 2011. Glacial to Holocene swings of the Australian-Indonesian monsoon. Nature Geoscience 4, 540-544.

Finally, a recent study suggests that annual mean rainfall during the LGM was even higher than today in the western part of the Australian-Indoneian monsoon system using different proxies and models. Reference: Mohtadi M., Prange M., Schefuß E., Jennerjahn T.C., 2017. Late Holocene Slowdown of the Indian Ocean Walker circulation. Nature Communications 8, doi:10.1038/s41467-017-00855-3.

Finally, additional support for the above inferences is provided by Niedermeyer et al. (2014), showing no difference in LGM rainfall compared to today. Reference: Niedermeyer E.M., Sessions A.L., Feakins S.J., Mohtadi M., 2014. Hydroclimate of the western Indo-Pacific Warm Pool during the past 24,000 years. Proceedings of the National Academy of Science 111, 9402-9406.

I can hardly imagine how the authors have missed these papers, all being published in top-ranked journals, and why the above results are not discussed in this study, as they are of great relevance for both model and proxy evaluations. I look forward to see those results implemented in the discussion section.

Best regards, Mahyar Mohtadi

---

## Referee Comment (RC1) · Anonymous Referee #1 · 22 May 2018

General Comments

This paper presents a multi-model study of changes in the Australian monsoon at the Last Glacial Maximum based on PMIP3 simulations. The topic is an important one, and the study presents interesting results showing a change in seasonality of rainfall, with increased seasonal cycle due to winter drying and early summer rainfall increases. The mechanisms producing this change are also explored, and the role of local circulation changes due to altered land configuration is identified as a major contribution to the changes.

The study fails to adequately introduce the climate models used, or to deal with

the uncertainty due to model biases or model disagreement on the sign of rainfall changes. The study also employs an overly simplistic method to decompose quantitative changes in rainfall due to dynamic and thermodynamic factors, relying entirely on multi-model mean changes and using a Clausius-Clapeyron scaling that is too large for the thermodynamic component.

Despite these limitations, I believe the study could make a valuable contribution to our understanding of Australian monsoon rainfall changes under LGM conditions. Major revisions are recommended, as outlined below in my comments.

Specific Comments

1. Line 69: The Australian monsoon is not defined clearly here or elsewhere, and the definition is not consistent throughout the paper or with other studies. Which domain is used? Does it include the Maritime Continent? Are land and ocean model grid points used? Is the domain the same in all models? How is the area shown in red in Figure 1a defined, and why does it include parts of the South Pacific Convergence Zone?

Note that the largest rainfall changes (Fig 1a) are over ocean to the north of Australia. If the results in this study are the area average over the grid points enclosed by the red line, then they represent mainly changes over PNG and the Maritime Continent, which makes it difficult to compare with proxy records or model studies focused on Northern Australian land areas. I suggest to re-calculate rainfall changes over Northern Australian land areas only (e.g. to 20S or 25S) and discuss and consider whether the results are consistent with those for the larger Australia-Maritime Continent domain.

Also, monsoon strength or intensity is defined in several different ways. Here (line 70) it is stated that a strong monsoon means wet conditions, whereas elsewhere a strong or intense monsoon means a large seasonal difference in rainfall between wet and dry seasons.

Please clarify: What is the monsoon domain used? Does it include both land and

ocean? How is monsoon strength and intensity defined?

2. Lines 72-75: Several of these records are not from the monsoon region, so are not relevant here.

3. Line 94: Multi-model ensembles can also provide a clearer perspective on model uncertainty (when all models agree, the result may be more robust – although not always, as models may share systematic biases).

4. Lines 102-104: "This result... has not been proved yet" – it is not clear whether this discussion refers to models or proxy records. It is important to distinguish between these two sources of information, and to acknowledge that neither provides a "true" record of the LGM as proxy records require interpretation and calibration and may be spatially incomplete, while models contain biases.

5. Line 104-107: Bayon et al. (2017) discussion of subtropics is not referring to the monsoon, which lies within the tropics. Remove or modify this sentence.

6. Line 122: How many models were used? Comment on the model skill in simulating the Australian monsoon rainfall: the models used in PMIP3 are typically lower resolution CMIP5 models, and many do not have high skill in simulating regional rainfall. At least, cite some model evaluation studies of the Australian monsoon in CMIP5 models, e.g. Jourdain et al. (2013), Brown et al. (2016) and summarise model skill in this region.

7. Line 166: According to Held and Soden (2006), who should be cited here, global precipitation would be expected to increase (or decrease) by around 2%/K. Previous studies have found a slightly higher scaling of around 3%/K for Asian monsoon rainfall (Endo and Kitoh, 2014).

8. Page 8, first paragraph: I am not comfortable with a quantitative decomposition based on the multi-model mean. The sign and magnitude of changes will be different in each model and the decomposition is only valid for individual models. Also, the scaling

of precipitation with temperature is likely too strong (see point above). Further, can all these changes be considered linearly? A more robust decomposition of dynamic and thermodynamic changes in each model should be applied, e.g. Seager et al. (2010), Chadwick et al. (2013) or Endo and Kitoh (2014).

9. Line 241: Where do the monsoon percentage changes come from? The rainfall changes in November-December shown in Figure 6 are in mm/day not %. The model spread (agreement) should also be discussed here and elsewhere: how many models simulate increased rainfall in the LGM and how many simulate decreased rainfall? How does this influence our confidence in the MMM changes?

10. Line 247: See point 1 above, please use a consistent definition of monsoon intensity. I suggest use "intensified seasonality" here for clarity. It is also necessary to describe in this paper how the average summer or wet season rainfall changes at the LGM, as this is the normal measure of the strength of the Australian summer monsoon. You should show (e.g. in a bar chart or table) annual mean and wet season (November to April) rainfall change for EACH model and for the MMM. This provides the context for the more detailed discussion of changes in seasonality and is more directly comparable with proxy reconstructions of annual or wet season rainfall and with studies of future monsoon (wet season) rainfall changes.

11. Line 258-265: The discussion of Tharammal (2017) is confusing. Do your results agree with theirs? If so, then simply state this.

12. Line 278: Why would the precipitation change lag the insolation change by two months? Provide a reference.

13. Line 288: "Strong convergence rain belt": Do you mean the ITCZ?

14. Line 291: A little more northerly? It is not clear what is being compared to what here.

15. Line 309 and line 314: See discussion under point 10. State that the monsoon

seasonality is amplified or intensified (rather than the monsoon itself).

16: Page 12, paragraphs 2 and 3: I repeat that I am not comfortable with a quantitative MMM decomposition. At least, you need to make it clear that your results are MMM values and state the model spread or uncertainty as well.

17. Figure 1: How is the monsoon domain defined? Why does it include the SPCZ region? Show some measure of model spread in Figure 1b, such as standard deviation of model range.

18. Figure 10: It may be more useful to show a smaller domain, excluding the North Pacific, with a smaller contour range. This would make the changes in Pacific and Indian Ocean tropical SSTs easier to see.

19. Figure 11: What is the "increased AR region" (11b)? What is the "central Australian monsoon region" referred to in the caption? Define the domain used.

20: Figure 12: I am not sure if this diagram is very useful. Also, arrows (if any) and linking lines are not clear in my print version.

21: Table 1: Were all model run years used from each model? This should be mentioned in Section 2. It would be more consistent to use the same number of years from each model.

Technical Corrections

Line 85: Change wording: "The change in the Australian monsoon was inconclusive..."

Line 94: Multi-model ensembles can reduce or cancel out the biases, not "delineate" (describe, define) them.

Line 107: Remove "insight" before "studies".

Line 110: Here and elsewhere in the paper, use "thermodynamic" not "thermal dynamic".

Line 127: A simpler version of the PMIP3 website address is: https://pmip3.lsce.ipsl.fr/.

Line 151: Here and elsewhere, do not use the American term "Fall" to refer to Southern Hemisphere Autumn (use "Autumn").

Line 159: Insert "global" before "temperature and humidity".

Line 205: Remove "We noticed that".

Line 256: It is not clear what the personal communication refers to here, I suggest remove it.

Line 307: Insert "global mean" before "temperature and water vapor".

Line 336: "Synthesized" does not make sense: should this be "simulated" (i.e. from models) or "multi-model mean" (i.e. averaged over many models)?

Line 469: Treble reference is incorrectly appended to Tharammal reference.

References: Brown, J.R, A. F. Moise, R. Colman, and H. Zhang (2016), Will a warmer world mean a wetter or drier Australian monsoon? J. Clim., 29, 4577-4596, doi:10.1175/JCLI-D-15-0695.1.

Chadwick, R., I. Boutle, and G. Martin, 2013: Spatial patterns of precipitation change in CMIP5: Why the rich do not get richer in the tropics. J. Climate, 26, 3803–3822, doi:10.1175/JCLI-D-12-00543.1.

Endo, H., and A. Kitoh, 2014: Thermodynamic and dynamic effects on regional monsoon rainfall changes in a warmer climate. Geophys. Res. Lett., 41, 1704–1710, doi:10.1002/2013GL059158.

Held, I. M., and B. J. Soden, 2006: Robust responses of the hydrological cycle to global warming. J. Climate, 19, 5686–5699, doi:10.1175/JCLI3990.1.

Jourdain, N. C., A. Sen Gupta, A. S. Taschetto, C. C. Ummenhofer, A. F. Moise, and K. Ashok, 2013: The Indo-Australian monsoon and its relationship to ENSO and IOD

in reanalysis data and the CMIP3/CMIP5 simulations. Climate Dyn., 41, 3073–3102, doi:10.1007/s00382-013-1676-1.

Seager, R., N. Naik, and G. Vecchi, 2010: Thermodynamic and dynamic mechanisms for large-scale changes in the hydrological cycle in response to global warming. J. Climate, 23, 4651–4668, doi:10.1175/2010JCLI3655.1.

Please also note the supplement to this comment:
https://www.clim-past-discuss.net/cp-2018-24/cp-2018-24-RC1-supplement.pdf

---

## Referee Comment (RC2) · Anonymous Referee #2 · 1 Jun 2018

The authors examined response of Australian monsoon to LGM forcing among CMIP5/PMIP3 multiple models. Simulated annual range of Australian monsoon rainfall during LGM is larger than present day, distinct from other regional monsoon systems. However, in a previous paper published in 2016, it has been already explored that this unique monsoon behavior was found among CMIP5/PMIP3 models and changes in land-sea contrast (due to change in land sea configuration arising from sea level drop) and east-west SST gradient are important for that. In that paper, the authors emphasized dynamic contribution to the spring-to-summer monsoon enhancement (rooted from changes in land-sea contrast and SST gradient) because thermodynamic contribution (reduced surface water vapor rooted from surface cooling) cannot explain

this enhancement. Most of the contents described in the current paper are just re-confirmations of previous paper (Yan et al. 2016).

In the current paper, the authors also tried to quantify relative contributions of dynamic and thermodynamic components related to the LGM Australian monsoon response. However, their quantitative decomposition is not reasonable. They did not follow widely-accepted methodology decomposing dynamic and thermodynamic components of rainfall response under climate change based on concepts of atmospheric water vapor budget. They also simply compared model-ensemble-mean anomaly between LGM and present day and dismissed inter-model differences in regional gradients in temperature, pressure and circulation response although they are essential for their main discussion. As an overall evaluation, novelty of this study seems very limited. I would like to recommend the authors to conduct any additional tests (e.g. Chiang et al. 2003: Toracinta et al. 2004; Ueda et al. 2011) to quantify effect of the land configuration (for example) to the Australian monsoon circulation and rainfall. Such sensitivity tests in addition to the quantitative evaluation of the hydrological response in multiple models are necessary for improving quality of this study.

Other comments

1. Please follow commonly-used dynamic-thermodynamic decomposition method. In line 165-173, 183-191 and other parts, ratio of specific humidity change should not be simply converted to that of precipitation change. Please read carefully Held and Soden 2006, O'Gorman et al. 2012 to catch current understanding of response of hydrological cycle under climate change, and Chou et al. 2009, Seager et al. 2010, and Chadwick et al. 2013 to understand widely-accepted methods for decomposition of dynamic and thermodynamic contributions to rainfall response under different climate states.

2. Please show inter-model consistency in (1) regional gradient in surface temperature, sea level pressure and rainfall, and (2) east-west SST gradient. In this paper, the authors checked inter-model consistency in LGM anomaly compared to PI. However, inter-model consistencies in the regional gradients in LGM anomaly (for example, are east-west dSST gradients really consistent among 7 models?) are not accessed although they are essential for the conclusion.

3. Please check inter-model consistency in LGM land configuration. Although the LGM land configuration was specified in PMIP3 protocol, land configuration implemented in each model could be different because model resolutions are much different between different model. Land-sea mask data in native grid of each model should be checked because any inter-model difference possibly affect inter-model difference in results.

4. Figures S1 and S2 seem identical to Figures 2 and 1 of Yan et al. (2016). You may need any copyright permission from Springer-Nature.

5. Line 26: relative -> related?

6. Line 41-44: I couldn't catch what do you mean here. Are "the local processes" you mention here land-sea configurations?

7. Line 110: thermal dynamics -> thermodynamic

References Chadwick, R., O. Boutle, and G. Martin (2013), Spatial patterns of precipitation change in CMIP5: Why the rich do not get richer in the tropics, J. Clim., 26, 3803-3822. Chiang, J. C., M. Biasutti, and D. S. Battisti (2003), Sensitivity of the Atlantic Intertropical Convergence Zone to Last Glacial Maximum boundary conditions, Paleoceanography, 18(4), 1094, doi:10.1029/2003PA000916. Chou, C., J. D. Neelin, C.-A. Chen, and J.-Y. Tu, 2009: Evaluating the "rich-get-richer" mechanism in tropical precipitation change under global warming. J. Climate, 22, 1982-2005. Held, I. M., and B. J. Soden, 2006: Robust responses of the hydrological cycle to global warming. J. Climate, 19, 5686-5699. O'Gorman PA, Allan RP, Byrne MP, Previdi M (2012) Energetic constraints on precipitation under climate change. Surv Geophys 33:585-608. Seager, R., N. Naik, and G. A. Vecchi, 2010: Thermodynamic and dynamic mechanisms for large-scale changes in the hydrological cycle in response to global

warming. J. Climate, 23, 4651-4668. Toracinta ER, Oglesby RJ, Bromwich DH (2004) Atmospheric response to modified CLIMAP ocean boundary conditions under the last glacial maximum. J Climate 17:504-522 Ueda, H., H. Kuroki, M. Ohba, and Y. Kamae, 2011: Seasonally asymmetric transition of the Asian monsoon in response to ice age boundary conditions. Clim. Dyn., 37, 2167-2179. Yan, M., Wang, B., and Liu, J. 2016: Global monsoon change during the Last Glacial Maximum: a multi-model study, Climate Dynamics, 47, 359-374.

---

## Author Comment (AC1) · 13 Aug 2018

General Comments This paper presents a multi-model study of changes in the Australian monsoon at the Last Glacial Maximum based on PMIP3 simulations. The topic is an important one, and the study presents interesting results showing a change in seasonality of rainfall, with increased seasonal cycle due to winter drying and early summer rainfall increases. The mechanisms producing this change are also explored, and the role of local circulation changes due to altered land configuration is identified as a major contribution to the changes. The study fails to adequately introduce the climate models used, or to deal with the uncertainty due to model biases or model disagreement on the sign of rainfall changes. The study also employs an overly simplistic method to decompose quantitative changes in rainfall due to dynamic and thermodynamic factors, relying entirely on multi-model mean changes and using a Clausius-Clapeyron scaling that is too large for the thermodynamic component. Despite these limitations, I believe the study could make a valuable contribution to our understanding of Australian monsoon rainfall changes under LGM conditions. Major revisions are recommended, as outlined below in my comments.

Reply to General Comments: Thank you very much for the comments on the models' uncertainties and on the method of attribution of the changes in precipitation to dynamic and thermodynamic factors. In the revised version, we implemented a brief discussion of the models' uncertainties along with the possible factors leading to the model biases, see Lines 383-405 in the revised text. We have also added a short description of the decomposition method in the revised version. Please find the details in the Reply to Specific Comment 8. We acknowledge your valuable comments and suggestions to improve our work.

Specific Comments 1. Line 69: The Australian monsoon is not defined clearly here or elsewhere, and the definition is not consistent throughout the paper or with other studies. Which domain is used? Does it include the Maritime Continent? Are land and ocean model grid points used? Is the domain the same in all models? How is the area shown in red in Figure 1a defined, and why does it include parts of the South Pacific Convergence Zone? Note that the largest rainfall changes (Fig 1a) are over ocean to the north of Australia. If the results in this study are the area average over the grid points enclosed by the red line, then they represent mainly changes over PNG and the Maritime Continent, which makes it difficult to compare with proxy records or model studies focused on Northern Australian land areas. I suggest to re-calculate rainfall changes over Northern Australian land areas only (e.g. to 20S or 25S) and discuss and consider whether the results are consistent with those for the larger Australia-Maritime Continent domain. Also, monsoon strength or intensity is defined in several

different ways. Here (line 70) it is stated that a strong monsoon means wet conditions, whereas elsewhere a strong or intense monsoon means a large seasonal difference in rainfall between wet and dry seasons. Please clarify: What is the monsoon domain used? Does it include both land and ocean? How is monsoon strength and intensity defined?

Reply: The monsoon domain is defined following hydroclimate definition, i.e., a contrast between wet summer and dry winter (Wang and Ding 2008). The monsoon domain is defined by the area where the annual range (local summer minus local winter) exceeds 2.0 mm/day, and the local summer precipitation exceeds 55% of the annual total precipitation. Here in the southern hemisphere, summer means November to March and winter means May to September. Since the domains derived from different models are different, and the changes of domain are also different, we use the fixed domain derived from the merged CMAP-GPCP precipitation data. The domain includes both land and ocean areas. A brief statement has been added as the Sec. 2.3 in the revised version, Lines 173-184. Note that the monsoon domain is shown only to give a general view of precipitation change, but not the main focus of this study. The seasonal distribution of the area averaged precipitation (Fig. 1b) is not based on the monsoon domain, but on the area where the annual range is increased, which is (20°S-EQ, 115°E-145°E) (as seen in Fig. 1a). To make it easier to compare with proxy records, the rainfall changes are re-calculated over Northern Australian land areas (20°S-5°S, 120°E-145°E) as suggested. The monsoon intensity in this study is represented by the annual range or the seasonality, i.e., the local summer minus the local winter. This has been clarified throughout the revised version.

2. Lines 72-75: Several of these records are not from the monsoon region, so are not relevant here.

Reply: Those unrelated papers (Treble et al. 2017; Bowler et al. 2012) are removed from the revised version.

3. Line 94: Multi-model ensembles can also provide a clearer perspective on model uncertainty (when all models agree, the result may be more robust – although not always, as models may share systematic biases).

Reply: Good point. It has been added to the revised version, Lines 94-95.

4. Lines 102-104: "This result... has not been proved yet" – it is not clear whether this discussion refers to models or proxy records. It is important to distinguish between these two sources of information, and to acknowledge that neither provides a "true" record of the LGM as proxy records require interpretation and calibration and may be spatially incomplete, while models contain biases.

Reply: Thanks for your suggestion. "This result" refers to the "simulated results", changed in the revision, Line 102. Also, "Neither model outputs nor proxy records provide a "true" record of the LGM, as proxy records require interpretation and calibration and may be spatially incomplete, while models contain biases" has been added in the revised text, Lines 104-106.

5. Line 104-107: Bayon et al. (2017) discussion of subtropics is not referring to the monsoon, which lies within the tropics. Remove or modify this sentence.

Reply: Modified, Lines 108-110.

6. Line 122: How many models were used? Comment on the model skill in simulating the Australian monsoon rainfall: the models used in PMIP3 are typically lower resolution CMIP5 models, and many do not have high skill in simulating regional rainfall. At least, cite some model evaluation studies of the Australian monsoon in CMIP5 models, e.g. Jourdain et al. (2013), Brown et al. (2016) and summarise model skill in this region.

Reply: Seven models are used in this study (Table 1), four of which have a higher resolution than 2 degrees in atmospheric component. For the oceanic component, the resolutions are even higher in six models (except IPSL). The resolutions of the oceanic

components of each model have been added into Table 1 in the revised version. The suggested references have been added to illustrate the model performance. These models' performance in the Australian monsoon region has been summarized in the revised text, see Lines 129-132.

7. Line 166: According to Held and Soden (2006), who should be cited here, global precipitation would be expected to increase (or decrease) by around 2%/K. Previous studies have found a slightly higher scaling of around 3%/K for Asian monsoon rainfall (Endo and Kitoh, 2014).

Reply: Yes, you are right. Here the 7 % change per degree of temperature change comes from the Clausius-Clapeyron (CC) relation, which is also suggested by Held and Soden (2006).

8. Page 8, first paragraph: I am not comfortable with a quantitative decomposition based on the multi-model mean. The sign and magnitude of changes will be different in each model and the decomposition is only valid for individual models. Also, the scaling of precipitation with temperature is likely too strong (see point above). Further, can all these changes be considered linearly? A more robust decomposition of dynamic and thermodynamic changes in each model should be applied, e.g. Seager et al. (2010), Chadwick et al. (2013) or Endo and Kitoh (2014).

Reply: Yes, the changes are different in different models. However, we calculated the changes based on the areas where signal-to-noise ratio exceeds 1, which means this change is relatively robust among the models. Also, the 7 % /K ratio of global precipitation over temperature comes from the Clausius-Clapeyron (CC) relation, which is also suggested by Held and Soden (2006). The actual changes are nonlinear. But we have simplified the changes as linear. For attribution of precipitation changes, we use a simplified relation based on the linearized equation of moisture budget used in the previous works (Chou et al., 2003; Seager et al., 2010; Huang et al., 2013; Endo and Kitoh, 2014; Liu et al., 2016). Considering a quasi-equilibrium state, the vertical integrated

moisture conservation can be approximately written as $-\int_{1000}\nabla\cdot(q\vec{v})dp\approx=P-E$ (1) where q is specific humidity, $\vec{v}$ is horizontal velocity, p is pressure, P is precipitation, and E the surface evaporation. Since water vapor is concentrated in the lower troposphere, the vertical integrated total column moisture divergence can be approximately replaced by the integration from the surface to 500 hPa. Define the $\Delta$ (.) as the change from PI to the LGM, i.e.,

$\Delta(.)=(.)LGM-(.)PI$ (2)

Then the precipitation change $\Delta P$ can be approximately calculated as follows: $\Delta P=-\int_{p1000}^{p500}\Delta(q\cdot\nabla\vec{v})dp-\int_{p1000}^{p500}\Delta(\vec{v}\cdot\nabla q)\,dp+\Delta E$ (3) To further simplify the equation, we use $-\omega\_500$ to represent vertical integrated $\nabla\vec{v}$, and q at the surface to represent vertical integrated specific humidity (Huang et al., 2013). Thus, the precipitation change ($\Delta P$) can be represented as

$\Delta P\approx\omega\grave{\text{I}}\breve{\text{E}}\_500\cdot\Delta q+q\grave{\text{I}}\breve{\text{E}}\cdot\Delta\omega\_500+\Delta E-\Delta T\_adv$ (4)

where $\omega\,\grave{\text{I}}\breve{\text{E}}\_500$ is 500 hPa vertical velocity in PI, q $\grave{\text{I}}\breve{\text{E}}$ is surface specific humidity in PI, $\Delta T\_adv$ is the changes due to the moisture advection ($\int_{p}^{p500}\Delta(\vec{v}\cdot\nabla q)\,dp$). The first term in the right-hand side of (4) ($\omega\,\grave{\text{I}}\breve{\text{E}}\_500\cdot\Delta q$) represents thermodynamic effect (due to the change of q), and the second term (q $\grave{\text{I}}\breve{\text{E}}\cdot\Delta\omega\_500$) represents dynamic effect (due to the change of circulation). The above method has been added in the revised Sec. 2.2, Lines 151-172. The spatial distributions of each term in JJA and ND have been provided in the revised version as supplementary figures (Figure S3 and Figure S6). The descriptions are added in the revised text, Lines 226-229 and Lines 301-305. It is clear that the dynamic effect plays more important role than the thermodynamic effect in the precipitation change over Australia and Maritime Continent. But this is not always true for other regions, such as South Africa and South America, where the thermodynamic and dynamic effects have comparable contributions. Based on the new decomposition method, we modified the statements about the contributions of thermodynamic and dynamic effects.

9. Line 241: Where do the monsoon percentage changes come from? The rainfall changes in November-December shown in Figure 6 are in mm/day not %. The model spread (agreement) should also be discussed here and elsewhere: how many models simulate increased rainfall in the LGM and how many simulate decreased rainfall? How does this influence our confidence in the MMM changes?

Reply: The percentage of precipitation change is calculated by the difference between precipitation in LGME and in piControl divided by the climatology in piControl, i.e. (PLGME – P piContrl)/PpiControl *100. Five models simulate increased local summer rainfall and the other two simulate decreased rainfall, please refer to Fig. 13 in the revised version. We have added the area averaged results derived from each model in Table 3 in the revised version, including annual mean, local summer mean and annual range. A short summary has been added in the revised text, Lines 202-208. Since most models are in agreement, we can rely on the MME results.

10. Line 247: See point 1 above, please use a consistent definition of monsoon intensity. I suggest use "intensified seasonality" here for clarity. It is also necessary to describe in this paper how the average summer or wet season rainfall changes at the LGM, as this is the normal measure of the strength of the Australian summer monsoon. You should show (e.g. in a bar chart or table) annual mean and wet season (November to April) rainfall change for EACH model and for the MMM. This provides the context for the more detailed discussion of changes in seasonality and is more directly comparable with proxy reconstructions of annual or wet season rainfall and with studies of future monsoon (wet season) rainfall changes.

Reply: Thank you for the valuable suggestion. The seasonality is used in the revised version to represent the monsoon intensity. As we can see from the seasonal distribution of precipitation changes derived from each model (Fig. 1c) that the largest increasing occurs in the early austral summer (ND), the simulated wet season (November to April) mean precipitation is decreased in all the models. Therefore, we take local summer (DJF) as wet season. The annual mean, DJF mean and annual range of precipitation changes derived from each model are listed in Table 3 in the revised version.

11. Line 258-265: The discussion of Tharammal (2017) is confusing. Do your results agree with theirs? If so, then simply state this.

Reply: Yes, our results are in agreement with theirs. The confusing part has been deleted in the revised version.

12. Line 278: Why would the precipitation change lag the insolation change by two months? Provide a reference.

Reply: In the annual variation, precipitation responds to the lower tropospheric moisture convergence. The moisture change depends on temperature change while the circulation change depends on surface temperature gradients change. The change of the surface temperature lags insolation changes because of the ocean and land surfaces have heat capacity (thermal inertial). In other words, insolation is a heating rate which equals to temperature change (tendency) but not the temperature itself. This has been added into the revised version, see Lines 335-340.

13. Line 288: "Strong convergence rain belt": Do you mean the ITCZ?

Reply: Yes, it is related to the ITCZ. "ITCZ" is added in the revised text, Line 350.

14. Line 291: A little more northerly? It is not clear what is being compared to what here.

Reply: It is compared to the position in our study. We have clarified in the revised text, Line 352.

15. Line 309 and line 314: See discussion under point 10. State that the monsoon seasonality is amplified or intensified (rather than the monsoon itself).

Reply: Thank you again for point out the misleading statement. All have been changed into "seasonality" in the revised version. For example, Line 410 and Line 416.

16: Page 12, paragraphs 2 and 3: I repeat that I am not comfortable with a quantitative MMM decomposition. At least, you need to make it clear that your results are MMM values and state the model spread or uncertainty as well.

Reply: Changed in the revised version. The quantitative values have been removed. The model uncertainties have been concluded in Lines 443-445 in the revised text.

17. Figure 1: How is the monsoon domain defined? Why does it include the SPCZ region? Show some measure of model spread in Figure 1b, such as standard deviation of model range.

Reply: The monsoon domain is defined following Wang and Ding (2008), i.e., the areas where the annual range (local summer minus local winter) exceed 2.0 mm/day, and the ratio of local summer against annual mean precipitation exceeds 55 %. The definition has been added in the Sec. 2.3 in the revised version. The seasonal distribution derived from each model is added as Figure 1c in the revised version.

18. Figure 10: It may be more useful to show a smaller domain, excluding the North Pacific, with a smaller contour range. This would make the changes in Pacific and Indian Ocean tropical SSTs easier to see.

Reply: Actually, the SST change in a smaller domain was shown in Figure S4 in the original manuscript. SST change in a smaller domain has been used as Figure 10 in the revised version.

19. Figure 11: What is the "increased AR region" (11b)? What is the "central Australian monsoon region" referred to in the caption? Define the domain used.

Reply: The "increased AR region" is the "central Australian monsoon region", as shown in Fig. 1a. The region used in the revised version has been changed into North Australian land area as suggested, which is (20°S-5°S, 120°E-145°E). The figure caption has been modified in the revised version.

20: Figure 12: I am not sure if this diagram is very useful. Also, arrows (if any) and

linking lines are not clear in my print version.

Reply: In the revised version, we only show the local dynamic processes in the two seasons (JJA and ND), which we think is useful for understanding the mechanisms of precipitation change. We have used thick arrows to make it clearer. The modified figure is shown in Figure 16 in the revised version.

21: Table 1: Were all model run years used from each model? This should be mentioned in Section 2. It would be more consistent to use the same number of years from each model.

Reply: Yes, we have used the same number of years for each model to get the model climatology. It has been stated in Sec. 2.1 in the revised text, Lines 133-134.

Technical Corrections Line 85: Change wording: "The change in the Australian monsoon was inconclusive. . ."

Reply: Changed in the revised version, Line 84.

Line 94: Multi-model ensembles can reduce or cancel out the biases, not "delineate" (describe, define) them.

Reply: Modified in the revised version, Line 93.

Line 107: Remove "insight" before "studies".

Reply: Removed in the revised version, Line 107.

Line 110: Here and elsewhere in the paper, use "thermodynamic" not "thermal dynamic".

Reply: Yes, all have been fixed in the revised version.

Line 127: A simpler version of the PMIP3 website address is: https://pmip3.lsce.ipsl.fr/.

Reply: Yes, fixed, Line 138.

Line 151: Here and elsewhere, do not use the American term "Fall" to refer to Southern Hemisphere Autumn (use "Autumn").

Reply: All the "Fall" has been changed into "Autumn" in the revised version.

Line 159: Insert "global" before "temperature and humidity".

Reply: Yes, fixed. "global" has been added in the revised version, Line 211.

Line 205: Remove "We noticed that".

Reply: Yes, fixed, Line 230.

Line 256: It is not clear what the personal communication refers to here, I suggest remove it.

Reply: Removed in the revised version, Line 319.

Line 307: Insert "global mean" before "temperature and water vapor".

Reply: Yes, fixed. "global mean" has been added in the revision, Line 408.

Line 336: "Synthesized" does not make sense: should this be "simulated" (i.e. from models) or "multi-model mean" (i.e. averaged over many models)?

Reply: Deleted in the revised version, Line 448.

Line 469: Treble reference is incorrectly appended to Tharammal reference.

Reply: Thank you for pointing out this mistake. This redundant reference has been deleted in the revised version, Line 612.

Please refere to the Supplement File for convenience.

Please also note the supplement to this comment:
https://www.clim-past-discuss.net/cp-2018-24/cp-2018-24-AC1-supplement.pdf

---

## Author Comment (AC2) · 13 Aug 2018

Anonymous Referee #2 The authors examined response of Australian monsoon to LGM forcing among CMIP5/PMIP3 multiple models. Simulated annual range of Australian monsoon rainfall during LGM is larger than present day, distinct from other regional monsoon systems. However, in a previous paper published in 2016, it has been already explored that this unique monsoon behavior was found among CMIP5/PMIP3 models and changes in land-sea contrast (due to change in land sea configuration arising from sea level drop) and east-west SST gradient are important for that. In that paper, the authors emphasized dynamic contribution

to the spring-to-summer monsoon enhancement (rooted from changes in land-sea contrast and SST gradient) because thermodynamic contribution (reduced surface water vapor rooted from surface cooling) cannot explain this enhancement. Most of the contents described in the current paper are just reconfirmations of previous paper (Yan et al. 2016). In the current paper, the authors also tried to quantify relative contributions of dynamic and thermodynamic components related to the LGM Australian monsoon response. However, their quantitative decomposition is not reasonable. They did not follow widely-accepted methodology decomposing dynamic and thermodynamic components of rainfall response under climate change based on concepts of atmospheric water vapor budget. They also simply compared model-ensemble-mean anomaly between LGM and present day and dismissed inter-model differences in regional gradients in temperature, pressure and circulation response although they are essential for their main discussion. As an overall evaluation, novelty of this study seems very limited. I would like to recommend the authors to conduct any additional tests (e.g. Chiang et al. 2003: Toracinta et al. 2004; Ueda et al. 2011) to quantify effect of the land configuration (for example) to the Australian monsoon circulation and rainfall. Such sensitivity tests in addition to the quantitative evaluation of the hydrological response in multiple models are necessary for improving quality of this study.

Reply: Thank you for your valuable and constructive comments for improving our study. In the revised version, we have added the decomposing method to assess the hydrological response and have added two additional simulations to test the effect of land-sea configuration on Australian monsoon. Please find the detailed method of quantitative assessment of the hydrological response in the Reply to Comment 1. The additional simulations have been added in the Discussion Section in the revised text. To isolate the impacts of land-sea configuration change, two experiments are conducted using a fully coupled earth system model (NESM v1, Cao et al., 2015). One is the PI control run designed the same as PMIP3 protocol, the other is the same as PI control run but with LGM land-sea configuration. The sensitive simulation illustrates that the local dynamical process induced by the land-sea configuration change is essential
to the Australian monsoon precipitation change. The additional simulated results are shown in Figure 12 in the revised version. The additional simulations and results are included in the Discussion Section, Lines 367-382.

Other comments

1. Please follow commonly-used dynamic-thermodynamic decomposition method. In line 165-173, 183-191 and other parts, ratio of specific humidity change should not be simply converted to that of precipitation change. Please read carefully Held and Soden 2006, O'Gorman et al. 2012 to catch current understanding of response of hydrological cycle under climate change, and Chou et al. 2009, Seager et al. 2010, and Chadwick et al. 2013 to understand widely-accepted methods for decomposition of dynamic and thermodynamic contributions to rainfall response under different climate states.

Reply: Thank you for your suggestion. In the revised work, we have made a rigorous quantitative analysis of the precipitation response to dynamic and thermodynamic factors. For attribution of precipitation changes, we use a simplified relation based on the linearized equation of moisture budget used in the previous works (Chou et al., 2003; Seager et al., 2010; Huang et al., 2013; Endo and Kitoh, 2014; Liu et al., 2016). Considering a quasi-equilibrium state, the vertical integrated moisture conservation can be approximately written as: $-\int\_100\nabla\cdot(qv\ \vec{S})dp\approx=P-E$ (1) where q is specific humidity, $v\ \vec{S}$ is horizontal velocity, p is pressure, P is precipitation, and E the surface evaporation. Since water vapor is concentrated in the lower troposphere, the vertical integrated total column moisture divergence can be approximately replaced by the integration from the surface to 500 hPa. Define the $\Delta$ (.) as the change from PI to the LGM, i.e.,

$\Delta(.)=(.)LGM-(.)PI$ (2)

Then the precipitation change $\Delta P$ can be approximately calculated as follows: $\Delta P=-\int\_p100\text{\textcircled{}}p500\Delta(q\cdot\vec{L}\vec{G}v\ \vec{S}\ )dp- \int\_p100\text{\textcircled{}}p500\Delta(v\ \vec{S}\cdot\vec{L}\vec{G}q)\ dp+\Delta E$ (3) To further simplify the equation, we use $-\omega\_500$ to represent vertical integrated $\vec{L}\vec{G}v\ \vec{S}$, and q

at the surface to represent vertical integrated specific humidity (Huang et al., 2013). Thus, the precipitation change ($\Delta$P) can be represented as $\Delta P \approx \bar{L}i\omega$ ÌĚ_500·$\Delta$q+q ÌĚ·$\Delta\omega$_500+$\Delta$E-$\Delta$T_adv (4) where $\omega$ ÌĚ_500 is 500 hPa vertical velocity in PI, q ÌĚ is surface specific humidity in PI, $\Delta$T_adv is the changes due to the moisture advection ($\int$ _p@p500$\Delta$(v âČŚ·âĹĞq) dp). The first term in the right-hand side of (4) ($\omega$ ÌĚ_500·$\Delta$q) represents thermodynamic effect (due to the change of q), and the second term (q ÌĚ·$\Delta\omega$_500) represents dynamic effect (due to the change of circulation). The above method has been added in the revised Sec. 2.2, Lines 151-172. The spatial distributions of each term in JJA and ND have been provided in the revised version as supplementary figures (Figure S3 and Figure S6). The descriptions are added in the revised text, Lines 226-229 and Lines 301-305. It is clear that the dynamic effect plays more important role than the thermodynamic effect in the precipitation change over Australia and Maritime Continent. But this is not always true for other regions, such as South Africa and South America, where the thermodynamic and dynamic effects have comparable contributions. Based on the new decomposition method, we modified the statements about the contributions of thermodynamic and dynamic effects.

2. Please show inter-model consistency in (1) regional gradient in surface temperature, sea level pressure and rainfall, and (2) east-west SST gradient. In this paper, the authors checked inter-model consistency in LGM anomaly compared to PI. However, inter-model consistencies in the regional gradients in LGM anomaly (for example, are east-west dSST gradients really consistent among 7 models?) are not accessed although they are essential for the conclusion.

Reply: The inter-model consistencies of regional gradient in temperature, SLP and SST have been provided in the revised version as supplementary (Figure S5). The east-west SST gradient (warm western tropic Pacific Ocean and cold eastern tropic Indian Ocean) is consistent among the models, please refer to the Figure S5e.

3. Please check inter-model consistency in LGM land configuration. Although the LGM land configuration was specified in PMIP3 protocol, land configuration implemented in

each model could be different because model resolutions are much different between different model. Land-sea mask data in native grid of each model should be checked because any inter-model difference possibly affect inter-model difference in results.

Reply: Although the resolutions of atmospheric component in each model are much different, four of the seven models have higher resolutions than 2-degree. For the oceanic component, most models (except IPSL-CM5A-LR) have higher resolutions than 1-degree. We have added the resolutions used in the oceanic component of the models in Table 1. It's hard to obtain the land-sea mask data from each model, here we use the climatology of SST in the LGM to illustrate the land-sea configuration in each model (Figure A). We are focusing on the tropical Indian Ocean and tropical west Pacific Ocean. Note that the resolution of 7MME is 2.5°*2.5°, lower than the individual models. The resolution of land configuration might not be the key question that will affect the results.

4. Figures S1 and S2 seem identical to Figures 2 and 1 of Yan et al. (2016). You may need any copyright permission from Springer-Nature.

Reply: The paper of Yan et al. (2016) has been purchased "Open Access" in Climate Dynamics. So, we don't need to obtain the copyright.

5. Line 26: relative -> related?

Reply: Yes, it should "be related", changed in the revised version. Please refer to Line 26 in the revised text.

6. Line 41-44: I couldn't catch what do you mean here. Are "the local processes" you mention here land-sea configurations?

Reply: Some synthesis suggests that the change of Australian monsoon during the LGM might be related to the large-scale circulation change such as the shifted position of ITCZ. However, in this work we find that it is not closely related to this large-scale circulation change, but to the local dynamics. In this study, "the local dynamics" not only

represents the dynamics due to the "land-sea contrast", but also due to the "asymmetric SST changes between the east tropical Indian Ocean and tropical western Pacific Ocean". The statement has been modified in the revised version as follows: "The enhanced Australian monsoonality in the LGM is not associated with global scale circulation change such as the shift of the ITCZ, rather, it is mainly due to the change of regional circulations around Australia arising from the changes in land-sea contrast and the east-west SST gradients over the Indo-western Pacific oceans. This finding should be taken into account . . ." Please refer to Lines 42-45.

7. Line 110: thermal dynamics -> thermodynamic

Reply: Thank you for pointing out this. All the terms of "thermal dynamics" have been changed into "thermodynamics" in the revised version.

Please refer to the Supplement File for convenience.

Please also note the supplement to this comment:
https://www.clim-past-discuss.net/cp-2018-24/cp-2018-24-AC2-supplement.pdf
* * *
[Figure]

[Figure]

Figure A ND mean SST in LGME derived from each model and 7MME.

**Fig. 1.** Figure A ND mean SST in LGME derived from each model and 7MME

---

## Author Comment (AC3) · 13 Aug 2018

Dear Dr. Mohtadi,

On behalf of all co-authors, I really appreciate your comments.

In this work, we are investigating the seasonality of the Australian monsoon, not only the seasonal mean or annual mean. However, the results you provided are very important ones we are looking for, as they provide important synthesis for evaluating our results. We will add those studies in our discussion part. Thank you very much!

Best regards, Mi Yan

---

## Referee Report (RR1)

**Review of revised manuscript: "Understanding the Australian monsoon change during the Last Glacial Maximum with multi-model ensemble" by Yan et al.**

**General comments:**

The authors have gone to some effort to address reviewer comments on the manuscript. The revised manuscript now includes a more satisfactory decomposition of the rainfall changes based on Huang et al. (2013) approach using changes in specific humidity and vertical motion. The paper also now addresses the extent to which models agree on changes as well as presenting multi-model mean results.

I therefore recommend publication subject to minor corrections outlined below. (Please note that the manuscript would benefit from further proof-reading as I have not provided an exhaustive list of corrections.)

**Specific Comments:**

1. Line 27: replace "multi-models' experiments" with "multi-model experiments"

2. Line 34: delete "the" before northern Australia.

3. Line 130: while CMIP5 models have reasonable performance simulating the Australian monsoon precipitation, there are some biases, e.g. some models are much too wet or dry, or fail to simulate the reversal of winds. This is discussed in the Jourdain (2013) and Brown (2016) papers cited here. Please provide a more balanced assessment of model skill in simulating the Australian monsoon.

4. Line 168: Huang et al. (2013) argue that Tadv term can be neglected in the tropics to simplify the analysis. Is there a reason not to neglect this term?

5. Line 183-184: I am not sure what this sentence means. Are you arguing that the exact definition of the monsoon domain is not important for the results? Or that the analysis presented in the rest of the paper does not use this domain?

6. Line 226: "rest terms"? do you mean "the rest of the terms"?

7. Line 229 (Figure S3): Why do the terms not sum to give the total rainfall change? Also, note comment 4 above about neglecting the advection term.

8. Line 298: "Thermal effects" should be "thermodynamic effects" (here and in other places, e.g. lines 324, 414, 415).

9. Line 299: Replace "impact" with "magnitude" (i.e. quantitative comparison).

10. Line 309: Remove brackets?

11. Line 314: Replace "which is consists with our work in this point of view" with "which is also consistent with our work".

12. Line 316: Remind the reader what is shown by the reconstruction of Liu et al. (2015).

13. Line 330-351: This paragraph is a bit confusing. Perhaps remove some of the detail, e.g. lines 338-346. Also it needs careful proof reading for both English and for the scientific content of the discussion. Several different points are being mixed together, and the last two sentences (347-351) seem out of place.

14. Line 331: insert "relative to the present day" after November.

15. Line 334: delete "of" before "the ocean".

16. Line 365: replace "synthesis" with "hypothesis".

17. Line 378: replace "sensitive" with "sensitivity"

18. Line 379: replace "to be" with "are"

19. Line 385: replace "wind filed" with "wind field"

20. Line 395: replace East Pacific… patter" with "eastern Pacific El Nino-like pattern".

21. Lines 414, 415: thermodynamic not thermal

22. Line 425, 431: sensitivity not sensitive

23. Line 440-441: I would argue that this will not "improve model performance", but instead increase confidence in model results or understanding of model-data disagreement. Changes in the model physics, resolution etc. are required to improve model performance – a different matter entirely.

---

## Author Response (AR2)

**Suggestions for revision or reasons for rejection (will be published if the paper is accepted for final publication)**

In the revised manuscript, description of the experiments, analysis methods, and discussion are much improved from original version. I appreciate the authors' great effort for improving their draft according to reviewers' comments. I only have a few minor comments.

Reply: Thank you for your affirmation. We appreciate your constructive comments to improve our work.

1. Figure 13 is the key for mentioning inter-model difference in land configuration (land-sea mask) and simulated results. In Figure A of "response to reviewers", the authors showed each model's land/ocean mask. This should also be shown in Figure 13. I mean, coastlines shown in Fig. 13 should be consistent with Fig. A. I found that land area fraction data, sftlf, is available at CERA website.
https://cera-www.dkrz.de/WDCC/ui/cerasearch/q?page=0&query=sftlf+lgm&rows=15
In addition, please include Figure A into supplementary information. This is really important for mentioning inter-model consistency/difference in LGM SST gradient.

Reply: Thank you very much for providing the information of the dataset. We downloaded the land area fraction data and used in Fig. 13 for each model. And Fig. A is added as Fig. S9 in the revised version.

2. The description of model and sensitivity experiments explained in discussion section should be included in section 2. I mean, the subsection title of 2.1 should be modified (e.g. PMIP3 models and experiments), then new subsection (the NESM model experiments) should be added.

Reply: The new subsection has been added as Sec. "2.2 NESM model and experiments" in the revised version, and the section numbers have been changed accordingly.

3. >> Figures S1 and S2 seem identical to Figures 2 and 1 of Yan et al. (2016). You may need any copyright permission from Springer-Nature.
> Reply: The paper of Yan et al. (2016) has been purchased "Open Access" in Climate Dynamics. So, we don't need to obtain the copyright.
Even though it is open access, you should refer that paper in the figure caption. E.g. "reprint of Fig. XX of Yan et al. (2016)."

Reply: Thank you for pointing out this. We have added the reference in the figure caption of Figure S1 and S2.

**Suggestions for revision or reasons for rejection (will be published if the paper is accepted for final publication)**

Review of revised manuscript: "Understanding the Australian monsoon change during the Last Glacial Maximum with multi-model ensemble" by Yan et al.

General comments:

The authors have gone to some effort to address reviewer comments on the manuscript. The revised manuscript now includes a more satisfactory decomposition of the rainfall changes based on Huang et al. (2013) approach using changes in specific humidity and vertical motion. The paper also now addresses the extent to which models agree on changes as well as presenting multi-model mean results.

I therefore recommend publication subject to minor corrections outlined below. (Please note that the manuscript would benefit from further proof-reading as I have not provided an exhaustive list of corrections.)

Reply: Thank you for your affirmation and careful corrections. We have gone through carefully and made corrections.

Specific Comments:

1. Line 27: replace "multi-models' experiments" with "multi-model experiments"

Reply: Changed. Line 27.

2. Line 34: delete "the" before northern Australia.

Reply: Deleted. Line 34.

3. Line 130: while CMIP5 models have reasonable performance simulating the Australian monsoon precipitation, there are some biases, e.g. some models are much too wet or dry, or fail to simulate the reversal of winds. This is discussed in the Jourdain (2013) and Brown (2016) papers cited here. Please provide a more balanced assessment of model skill in simulating the Australian monsoon.

Reply: We have changed the statement in the revised version. Lines 147-152.

4. Line 168: Huang et al. (2013) argue that Tadv term can be neglected in the tropics to simplify the analysis. Is there a reason not to neglect this term?

Reply: The Tadv term listed here comes from the EQ. (3), we didn't make further simplification. But yes, actually this term could be negligible as seen in the Fig. S3 and Fig. S6.

5. Line 183-184: I am not sure what this sentence means. Are you arguing that the exact definition of the monsoon domain is not important for the results? Or that the analysis presented in the rest of the paper does not use this domain?

Reply: We mean that the analysis presented in the rest of the paper does not consider this domain. We added a statement in the revised version to make it easy to understand. Line 194.

6. Line 226: "rest terms"? do you mean "the rest of the terms"?

Reply: Yes, it means "the rest terms of the equation (4)". Changed in the revised version. Line 236.

7. Line 229 (Figure S3): Why do the terms not sum to give the total rainfall change? Also, note comment 4 above about neglecting the advection term.

Reply: The actual precipitation changes are nonlinear, so the linear sum of the terms might not meet the actual change. And yes, we confirm that the advection term is negligible.

8. Line 298: "Thermal effects" should be "thermodynamic effects" (here and in other places, e.g. lines 324, 414, 415).

Reply: All changed in the revised version. Lines 267, 308, 336, 424, 426.

9. Line 299: Replace "impact" with "magnitude" (i.e. quantitative comparison).

Reply: Changed.   Line 309.

10. Line 309: Remove brackets?

Reply: Yes, thank you for pointing out this mistake. Changed in the revised version.   Line 319.

11. Line 314: Replace "which is consists with our work in this point of view" with "which is also consistent with our work".

Reply: Changed. Line 325.

12. Line 316: Remind the reader what is shown by the reconstruction of Liu et al. (2015).

Reply: Yes, added in the revised version. Lines 327-328.

13. Line 330-351: This paragraph is a bit confusing. Perhaps remove some of the detail, e.g. lines 338-346. Also it needs careful proof reading for both English and for the scientific content of the discussion. Several different points are being mixed together, and the last two sentences (347-351) seem out of place.

Reply: We are trying to discuss the effect of the orbital induced insolation changes, which has different impacts on different hemisphere due to the different heat capacity of the land and the ocean. And find that the insolation change is not the main forcing that influences Australian monsoon precipitation change. Yes, some details can be removed. Lines 342-362.

14. Line 331: insert "relative to the present day" after November.

Reply: Added in the revised version. Line 343.

15. Line 334: delete "of" before "the ocean".

Reply: Deleted. Line 347.

16. Line 365: replace "synthesis" with "hypothesis".
Reply: Changed. Line 376.

17. Line 378: replace "sensitive" with "sensitivity"
Reply: The related "sensitive experiment" have all been changed with "sensitivity" in the revised version. Lines 86, 154, 158, 385, 436, 442.

18. Line 379: replace "to be" with "are"
Reply: Changed. Line 386.

19. Line 385: replace "wind filed" with "wind field"
Reply: Changed. Line 393.

20. Line 395: replace "East Pacific… patter" with "eastern Pacific El Nino-like pattern".
Reply: Changed. Line 403.

21. Lines 414, 415: thermodynamic not thermal
Reply: All changed in the revised version. Lines 267, 308, 336, 424, 426.

22. Line 425, 431: sensitivity not sensitive
Reply: All changed. Lines 86, 154, 158, 385, 436, 442.

23. Line 440-441: I would argue that this will not "improve model performance", but instead increase confidence in model results or understanding of model-data disagreement. Changes in the model physics, resolution etc. are required to improve model performance – a different matter entirely.
Reply: Yes, you are right. We have changed the statement. Lines 451-452.

[revised manuscript text omitted]

(a) JJA mean z850 with wind850

(b) ND mean z850 with wind850

72 74 76 78 80 82 84 86   m

**Figure S4** The (a) JJA mean, and (b) ND mean 850 hPa geopotential height (shading) with 850

hPa wind (vector) difference between LGME and piControl. The red lines enclose the monsoon domains. The thick black lines denote the coastal lines in LGME provided by CMIP5/PMIP3, and the thin black lines denote the coastal lines in piControl. Only those areas where signal-to-noise ratio exceeds one are plotted.

[Figure]

**Figure S5** The ND mean zonal (top panels) and meridional (bottom panels) gradient difference between LGME and piControl. (a)-(b) for surface air temperature anomaly, (c)-(d) for sea level pressure anomaly, and (e)-(f) for sea surface temperature anomaly. The thick black lines denote the coastal lines in LGME provided by CMIP5/PMIP3, and the thin black lines denote the coastal lines in piControl. Only those areas where signal-to-noise ratio exceeds one are plotted.

[Figure]

**Figure S6** Same as Fig. S3, but for ND mean.

[Figure]

**Figure S7** The ND mean SST difference between the LGME and piControl. The orange lines enclose the monsoon domains. Only those areas where signal-to-noise ratio exceeds one are plotted.

[Figure]

ND mean ITCZ location

**Figure S8** The 7MME ND mean ITCZ location defined by the meridional maximum precipitation between 20 °S and 20 °N. The red line indicates the position during the LGM and the blue line indicates the position during the PI.

[Figure]

**Figure S9** ND mean SST in LGME derived from each model and 7MME.

---

## Author Response (AR3)

**Editor Decision: Publish subject to technical corrections** (06 Dec 2018) by Pascale
Braconnot
Comments to the Author:
Dear authors,

I am happy to accept your manuscript for publication in climate of the past.
There are still some typos or sentence that would benefit from English cleaning. So I
recommend that you correct them to produce the final manuscript. A few examples I
found while reading the version of the manuscript highlighting the latest changes :
l 69 from clearly ---> from being clearly?
l 76, being a non native English myself, it is not clear to me that "in" is the right word
between drier and the LGM
l 134 illustrate ---> compute ?
-165 earth system ---> Earth system model
also check the English "which is designed the same as" doesn't seem to be correct
l 199 but not ---> but us not?
in addition monsoon should be added before domain
l 360 This indicates the insolation ---> this indicates that the insolation.

These are some examples, but please check everything.

Best regards
Pascale Braconnot

Author reply: Thank you very much for accepting our work.

There are still some typos or sentence that would benefit from English cleaning. So I
recommend that you correct them to produce the final manuscript. A few examples I
found while reading the version of the manuscript highlighting the latest changes :
l 69 from clearly ---> from being clearly?
Reply: Changed. Line 69.

l 76, being a non native English myself, it is not clear to me that "in" is the right word
between drier and the LGM
Reply: Yes, you are right, should be "during", changed. Line 76.

l 134 illustrate ---> compute ?
Reply: Changed into "calculate". Line 131.

-165 earth system ---> Earth system model
also check the English "which is designed the same as" doesn't seem to be correct
Reply: Modified in the revised version. Lines 155-162.

l 199 but not ---> but us not?
in addition monsoon should be added before domain
Reply: The statement has been changed. Lines 196-197.

l 360 This indicates the insolation ---> this indicates that the insolation.
Reply: Added in the revised version. Line 356.

These are some examples, but please check everything.
Reply: Thank you for pointing out this. We have checked the manuscript and made corrections. Lines 27, 55, 78, 131-132, 134, 139-141, 146, 170, 235, 250, 259, 267, 307, 315, 340, 345-346, 348, 361, 362, 366, 391, 404, 410, 422, 435, 442-443, 475.
The track version can be found in the following.

[revised manuscript text omitted]

